# Skyfall-GS: Synthesizing Immersive 3D Urban Scenes from Satellite Imagery

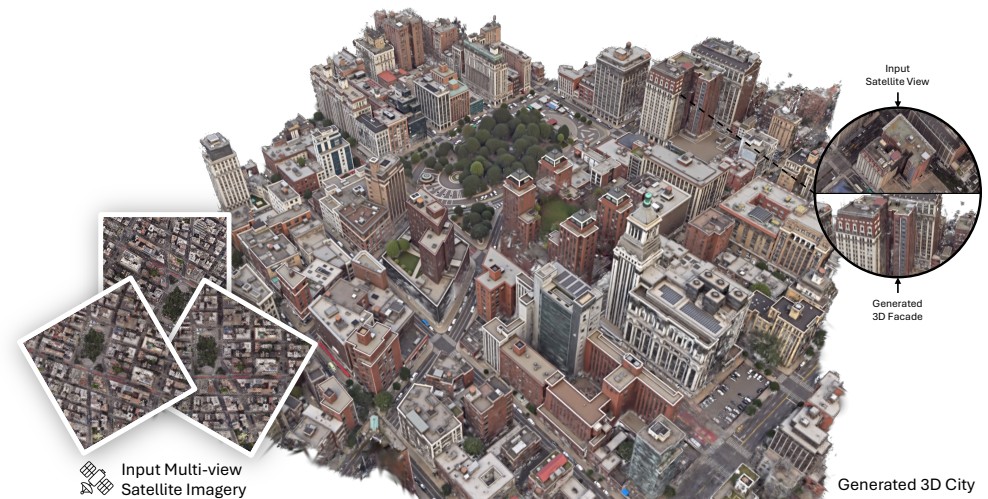

Figure 1: **Our method synthesizes high-quality, immersive 3D urban scenes solely from multi-view satellite imagery, enabling realistic drone-view navigation without relying on additional 3D or street-level training data.** Given multiple satellite images from diverse viewpoints and dates (*left*), our method leverages 3D Gaussian Splatting combined with pre-trained text-to-image diffusion models in an iterative refinement framework to generate realistic 3D block-scale city from limited satellite-view input (*right*). Our method significantly enhances visual fidelity, geometric sharpness, and semantic consistency, enabling real-time immersive exploration.

## Abstract

Synthesizing large-scale, explorable, and geometrically accurate 3D urban scenes is a challenging yet valuable task in providing immersive and embodied applications. The challenges lie in the lack of large-scale and high-quality real-world 3D scans for training generalizable generative models. In this paper, we take an alternative route to create large-scale 3D scenes by synergizing the readily available satellite imagery that supplies realistic coarse geometry and the open-domain diffusion model for creating high-quality close-up appearances. We propose **Skyfall-GS**, a novel hybrid framework that synthesizes immersive city-block scale 3D urban scenes by combining satellite reconstruction with diffusion refinement, eliminating the need for costly 3D annotations, also featuring real-time, immersive 3D exploration. We tailor a curriculum-driven iterative refinement strategy to progressively enhance geometric completeness and photorealistic textures. Extensive experiments demonstrate that Skyfall-GS provides improved cross-view consistent geometry and more realistic textures compared to state-of-the-art approaches.

## 1 Introduction

Synthetic high-quality, immersive, and semantically plausible 3D urban scenes have crucial applications in gaming, filmmaking, and robotics. The ability to create a large-scale and 3D-grounded environment supports realistic rendering and immersive experience for storytelling, demonstration, and embodied physics simulation. However, due to limited 3D-informed data, building a generative

model for realistic and navigable 3D cities remains challenging. It is expensive and labor-intensive to acquire large-scale 3D and textured reconstructions of cities with detailed geometry, while using Internet image collections face challenges in camera pose registration and excessive data noise (e.g., transient objects and different times of the day). These constraints set back existing 3D city generation frameworks from creating realistic and diverse appearances. With this observation, we propose an alternative route for virtual city creation with a two-stage pipeline: partial and coarse geometry reconstruction from multi-view satellite imagery, then close-up appearance completion and hallucination using an open-domain diffusion model.

Satellite imagery offers a compelling alternative due to its extensive geographic coverage, automated collection, and high-resolution capabilities. For instance, Maxar's WorldView-3 satellite captures approximately 680,000 km$^2$ of imagery daily at resolutions up to 31 cm per pixel. Such data inherently encodes semantically plausible representations of real-world environments, enabling scalable 3D urban scene creation. However, in Figure 2(a), we show that directly applying 3D reconstruction methods to satellite imagery is insufficient for creating *navigable and immersive* 3D cities. The significant amount of invisible regions (e.g., building facades) and limited satellite-view parallax create incorrect geometry and artifacts.

Completing and enhancing the geometry and texture in the ground view requires a significant influx of extra information. In Figure 2(b), we study a few state-of-the-art methods in city generation (Xie et al., 2024; 2025b). These methods produce oversimplified building geometries and unrealistic appearances due to strong assumptions, particularly the reliance on semantic maps and height fields as the sole inputs, and overfitting to small-scale, domain-specific datasets. Such an observation motivates us to seek help from open-domain foundation vision models as an external information source, which provides better zero-shot generalization and diversity. Noticing that the ground-view novel-view renderings from the GS reconstructed scene exhibit noise-like patterns, we treat these renderings as intermediate results in a denoising diffusion process. Then, we complete the remaining denoising process to create hallucinated pseudo ground-truth for the GS scene optimization. To stabilize the convergence, we carefully design a curriculum-based view selection and iterative refinement process, where the sampled view angles gradually *fall* from the *sky* to the ground over time. Accordingly, we name our framework **Skyfall-GS**. In Figure 1 and Figure 2, we show that Skyfall-GS yields significantly enhanced texture with 3D-justified geometry compared to the relevant baselines.

Skyfall-GS is a novel hybrid framework that synthesizes immersive city-block scale 3D urban scenes by combining satellite reconstruction with diffusion refinement, eliminating the need of fixed-domain training on 3D data. Skyfall-GS operates on readily available satellite imagery as the only input, then hallucinates realistic aerial-view appearances and maintains a strong satellite-to-ground 3D consistency. Moreover, Skyfall-GS supports real-time and interactive rendering, as we design our framework to produce GS results without sophisticated data structures. Through experiments on diverse environments, we show that Skyfall-GS has better generalization and robustness compared to state-of-the-art methods. Our ablation shows that each of our designs improves the perceptual plausibility and semantic consistency. Skyfall-GS paves the way for scalable 3D urban virtual scene creation, enabling applications in virtual entertainment, simulation, and robotics.

In summary, our contributions include:

- We introduce Skyfall-GS, the first method to synthesize immersive, real-time free-flight navigable 3D urban scenes solely from multi-view satellite imagery using generative refinement.
- An open-domain refinement approach leveraging pre-trained text-to-image diffusion models without domain-specific training.
- A curriculum-learning-based iterative refinement strategy progressively enhances reconstruction quality from higher to lower viewpoints, significantly improving visual fidelity in occluded areas.

## 2 RELATED WORK

**Gaussian Splatting.** 3D Gaussian Splatting (3DGS) (Kerbl et al., 2023) offers real-time view synthesis rivaling NeRFs (Mildenhall et al., 2021; Barron et al., 2021; 2022; Müller et al., 2022; Barron et al., 2023; Martin-Brualla et al., 2021). Mip-Splatting (Yu et al., 2024) fixes scale-change issues via on-the-fly resizing. Recent advances target satellite and aerial reconstruction: FusionRF (Sprintson et al., 2024) achieves 17% depth improvement from multispectral acquisitions, while InstantSplat (Fan et al., 2024) enables 40-second pose-free reconstruction. "In-the-wild" vari-

Figure 2: **Limitations of existing novel-view synthesis methods from satellite imagery.** (a) Sat-NeRF (Marí et al., 2022) and naive 3DGS (Kerbl et al., 2023) yield blurred or distorted building facades due to insufficient geometric detail and limited parallax from satellite viewpoints. (b) City generation methods (Xie et al., 2024; 2025b) produce oversimplified building geometries and unrealistic appearances, primarily due to strong assumptions about the input data, and overfitting to small-scale, domain-specific datasets. In comparison, our method synthesizes more realistic appearances and geometries from aerial views.

ants handle appearance and uncertainty (Xu et al., 2024; Sabour et al., 2024; Wang et al., 2024b; Dahmani et al., 2024; Zhang et al., 2024a; Kulhanek et al., 2024; Hou et al., 2025), including Spec-troMotion (Fan et al., 2025) for dynamic specular scenes, while large-scene methods use LOD and partitioning (Kerbl et al., 2024; Liu et al., 2025c; 2024; Lin et al., 2024; Turki et al., 2022; Tancik et al., 2022). CAT-3DGS (Zhan et al., 2025) achieves rate-distortion optimization via context-adaptive triplanes. For sparse-view satellite imagery, depth or co-regularization priors guide reconstruction (Li et al., 2024b; Zhang et al., 2024b; Zhu et al., 2023; Niemeyer et al., 2022; Lin et al., 2025), with SparseSat-NeRF (Zhang & Rupnik, 2023) adding dense depth supervision.

**Diffusion models for 3D reconstruction and editing.** Diffusion models (Rombach et al., 2022; Labs, 2024b) underpin image generation and editing. Early SDS pipelines DreamFusion (Poole et al., 2022) and Magic3D (Lin et al., 2023a) enabled text-to-3D, with ProlificDreamer (Wang et al., 2023) addressing over-smoothing via Variational Score Distillation. DreamGaussian (Tang et al., 2023) achieves 10x speedup via progressive densification, while GaussianDreamer (Yi et al., 2024) bridges 2D and 3D diffusion models. SDEdit (Meng et al., 2022), DDIM inversion (Mokady et al., 2022; Miyake et al., 2024), and FlowEdit (Kulikov et al., 2024) enable fine control. Extensions include sparse-view reconstruction (Wu et al., 2023; Liu et al., 2023b; Chen et al., 2024), with MVDream (Shi et al., 2023) enabling multi-view consistency. For 3D/4D generation (Gao et al., 2024b; Wu et al., 2024b; Melas-Kyriazi et al., 2024; Chung et al., 2023; Liu et al., 2023a) and scene editing (Haque et al., 2023; Wu et al., 2025; Ye et al., 2024b; Fang et al., 2024; Mirzaei et al., 2024; Dihlmann et al., 2024; Weber et al., 2024; Wu et al., 2024a; Wang et al., 2025), SPIn-NeRF (Mirzaei et al., 2023) handles occlusions via perceptual inpainting while CF-NeRF (Shen et al., 2022) provides uncertainty quantification. CorrFill (Liu et al., 2025a) enhances faithfulness via correspondence guidance, while AuraFusion360 (Wu et al., 2025) enables 360° scene inpainting for Gaussian Splatting. Instruct-NeRF2NeRF (Haque et al., 2023) refines NeRF views iteratively with Instruct-Pix2Pix (Brooks et al., 2023) for diffusion-driven 3D editing.

**Urban scene modeling.** Classic SfM-MVS pipelines extract DSMs from satellite pairs (Schönberger & Frahm, 2016; Zhang et al., 2019; Gao et al., 2023a), with MVS3D (Bosch et al., 2016) benchmarks for evaluation. Neural variants improve geometric fidelity (Derksen & Izzo, 2021; Marí et al., 2022; 2023; Zhou et al., 2024b; Leotta et al., 2019; Liu et al., 2025b; Qu & Deng, 2023; Gao et al., 2024a; Savant Aira et al., 2025; Huang et al., 2025), including Sat-NeRF (Marí et al., 2022), which utilizes NeRF for satellite imagery and SatMVS (Gao et al., 2021; 2023b) with RPC warping, yet both miss occluded facades. Generative synthesis divides into: (i) street-view methods (Li et al., 2024c; 2021; 2024d; Toker et al., 2021; Qian et al., 2023; Shi et al., 2022; Ze et al., 2025; Deng et al., 2024; Xu & Qin, 2025), including GeoDiffusion (Xiong et al., 2024) for mixed-view synthesis, Geospecific View Generation (Xu & Qin, 2024) achieving 10x resolution gains, and SkyDiffusion (Ye et al., 2024a) with Curved-BEV for street-to-satellite mapping, though lacking 3D consistency and temporal coherence; and (ii) full-3D city generation (Lin et al., 2023b; Xie et al., 2024; 2025a;b; Sun et al., 2024; Zhou et al., 2024a; Shang et al., 2024; Li et al., 2024a; Zhang et al., 2024c), with BEVFormer (Li et al., 2022) and MagicDrive (Gao et al., 2023c) using spatiotemporal transformers for view consistency. While Infinicity (Lin et al., 2023b) uses pixel-to-voxel rendering for infinite cities, and CityDreamer (Xie et al., 2024) and GaussianCity (Xie et al., 2025b) use BEV neural fields or BEV-Point splats for editable scenes, these remain constrained by input representations (semantic maps and height fields) and training distributions, limiting synthesis of realistic textures and complex structures like tunnels, bridges, and multi-level architectures. Our method uses pretrained diffusion priors to recover

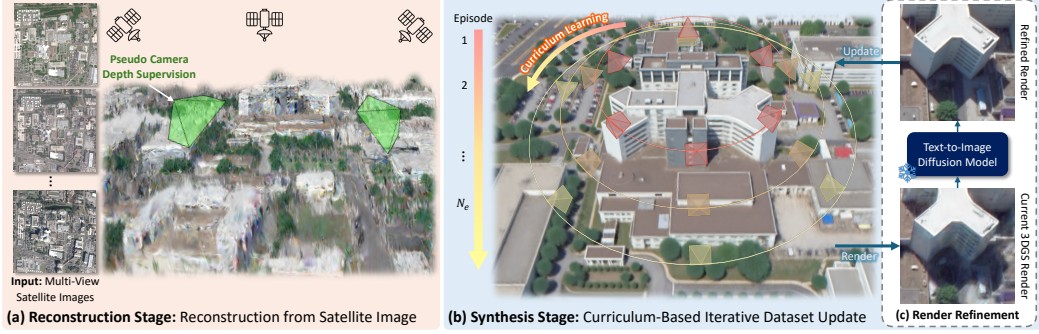

(a) **Reconstruction Stage:** Reconstruction from Satellite Image   (b) **Synthesis Stage:** Curriculum-Based Iterative Dataset Update   (c) **Render Refinement**

Figure 3: **Overview of the proposed Skyfall-GS pipeline.** Our method synthesizes immersive and free-flight navigable city-block scale 3D scenes solely from multi-view satellite imagery in two stages. (a) In the Reconstruction Stage, we first reconstruct the initial 3D scene using 3DGS, enhanced by pseudo-camera depth supervision to address limited parallax in satellite images. We use an appearance modeling component to handle varying illumination conditions across multi-date satellite images. (b) In the Synthesis Stage, we introduce a curriculum-based Iterative Dataset Update (IDU) refinement technique leveraging (c) a pre-trained T2I diffusion model (Labs, 2024b) with prompt-to-prompt editing (Kulikov et al., 2024). By iteratively updating training datasets with progressively refined renders, our approach significantly reduces visual artifacts, improving geometric accuracy and texture realism, particularly in previously occluded areas such as building facades.

high-fidelity facades in occluded regions without dataset-specific training, respecting user constraints more faithfully.

## 3 METHOD

Our two-stage pipeline (Figure 3) turns satellite images into immersive 3D cities. Reconstruction Stage (Section 3.1): fit a 3D Gaussian Splatting model, adding illumination-adaptive appearance modeling and regularizers for sparse, multi-date views. Synthesis Stage (Section 3.2): recover occluded regions, e.g., facades, through curriculum Iterative Dataset Update, repeatedly refining renders with text-guided diffusion edits. The loop keeps textures faithful to the satellite input while preserving geometry, yielding complete, navigable urban scenes from satellite data alone.

**Preliminary.** 3D Gaussian Splatting (3DGS) (Kerbl et al., 2023) encodes a scene as Gaussians with center $\mu_i$, covariance $\Sigma_i$, opacity $\alpha_i$, and view-dependent color. Each Gaussian projects to the image plane with covariance: $\Sigma_i' = JW\Sigma_i W^T J^T$, where $W$ is the viewing transformation and $J$ is the affine-projection Jacobian. Pixels are alpha-composited front-to-back. Parameters are trained with:

$$\mathcal{L}_{\text{color}} = \lambda_{\text{D-SSIM}} \, \text{DSSIM}(\hat{C}, C) + (1 - \lambda_{\text{D-SSIM}}) \|\hat{C} - C\|_1 \, . \tag{1}$$

### 3.1 INITIAL 3DGS RECONSTRUCTION FROM SATELLITE IMAGERY

The initial 3DGS reconstruction must faithfully preserve the texture and geometry of satellite imagery to provide a robust foundation for synthesis. We employ appearance modeling to handle variations in multi-date imagery. Since limited satellite parallax creates floating artifacts, we apply regularization techniques to constrain both texture and geometry.

**Approximated camera parameters.** Satellite imagery typically uses the rational polynomial camera (RPC) model, directly mapping image coordinates to geographic coordinates. To integrate with the 3DGS pipeline, we employ SatelliteSfM (Zhang et al., 2019) to approximate perspective camera parameters (extrinsic and intrinsic) from RPC and generate sparse SfM points as initial 3DGS points.

**Appearance modeling.** As highlighted in Section 1, multi-date satellite imagery exhibits significant appearance variations due to global illumination changes, seasonal factors, and transient objects, as illustrated in Figure 3(a). Following WildGaussians (Kulhanek et al., 2024), we use trainable per-image embeddings $\{e_j\}_{j=1}^N$ (with $N$ training images) to handle varying illumination and atmospheric conditions. We also employ trainable per-Gaussian embeddings $g_i$ to capture localized appearance changes like shadow variations. A lightweight MLP $f$ computes affine color transformation parameters $(\beta, \gamma)$ as $(\beta, \gamma) = f(e_j, g_i, \bar{c}_i)$, where $e_j$ is the per-image embedding, $g_i$ is the per-Gaussian embedding, and $\bar{c}_i$ denotes the 0-th order spherical harmonics (SH). Finally, the transformed color $\tilde{c}_i$

is then computed as $\tilde{c}_i(\mathbf{r}) = \gamma \cdot \hat{c}_i(\mathbf{r}) + \beta$, and used in the 3DGS rasterizer. To prevent modeling the appearance changes as view-dependent effects, we limit SH to zero and first-order terms.

**Opacity regularization.** We observed that numerous floaters in reconstructed scenes exhibit low opacity. To encourage geometry to adhere closely to actual surfaces, we propose entropy-based opacity regularization:

$$\mathcal{L}_{\text{op}} = -\sum_i \alpha_i \log(\alpha_i) + (1 - \alpha_i) \log(1 - \alpha_i) \ . \tag{2}$$

This regularization promotes binary opacity distributions, allowing low-opacity Gaussians to be pruned during densification. Incorporating this term significantly sharpens geometric reconstruction, providing a better foundation for subsequent synthesis.

**Pseudo camera depth supervision.** To further reduce floating artifacts, we sample pseudo-cameras positioned closer to the ground during optimization. From these pseudo-cameras, we render RGB images $I_{\text{RGB}}$ and corresponding alpha-blended depth maps $\hat{D}_{\text{GS}}$. We then use an off-the-shelf monocular depth estimator, MoGe (Wang et al., 2024a), to predict scale-invariant depths $\hat{D}_{\text{est}}$ from these renders. We use the absolute value of Pearson correlation (PCorr) to supervise the depth:

$$\mathcal{L}_{\text{depth}} = \|\text{PCorr}(\hat{D}_{\text{GS}}, \hat{D}_{\text{est}})\|_1 \ ; \quad \text{PCorr}(\hat{D}_{\text{GS}}, \hat{D}_{\text{est}}) = \frac{\text{Cov}(\hat{D}_{\text{GS}}, \hat{D}_{\text{est}})}{\sqrt{\text{Var}(\hat{D}_{\text{GS}})\text{Var}(\hat{D}_{\text{est}})}} \ . \tag{3}$$

**Optimization.** Combining all components, the overall loss for the reconstruction stage is defined as:

$$\mathcal{L}_{\text{sat}}(G, C) = \mathcal{L}_{\text{color}} + \lambda_{\text{op}}\mathcal{L}_{\text{op}} + \lambda_{\text{depth}}\mathcal{L}_{\text{depth}} \ , \tag{4}$$

where $G$ is the 3DGS representation, $C$ is the set of ground-truth satellite images, $\lambda_{\text{op}}$ and $\lambda_{\text{depth}}$ weight opacity regularization and depth supervision relative to the color reconstruction loss.

## 3.2 Synthesize via Curriculum-learning Based Iterative Datasets Update

The iterative dataset update (IDU) technique (Haque et al., 2023; Melas-Kyriazi et al., 2024) repeatedly executes render-edit-update cycles across multiple episodes to progressively synthesize 3D scenes. Unlike previous methods that sample camera poses from original training views (Haque et al., 2023) or simple orbits (Melas-Kyriazi et al., 2024), we introduce a curriculum-based refinement schedule over $N_e$ episodes that specifically addresses satellite imagery's geometric and visual limitations, producing structurally accurate and photorealistic reconstructions of occluded areas.

**Curriculum learning strategy.** As illustrated in Figure 4, we observe that 3DGS trained from satellite imagery produces higher-quality renders at higher elevation angles but degenerates at lower elevation angles. Leveraging this insight, we introduce a curriculum-based synthesizing strategy, which progressively lowering viewpoints across optimization episodes. Specifically, we define $N_p$ look-at points $\{P_i\}_{i=1}^{N_p}$ uniformly placed throughout the scene and uniformly sample $N_v$ camera positions along orbital trajectories with controlled elevation angles and radii. Our iterative dataset update (IDU) process starts from higher elevations, progressively moving toward lower perspectives. This approach gradually reveals previously occluded regions, improving geometric detail and texture realism, as validated in our ablation studies (Section 4.2).

**Render refinement by text-to-image diffusion model.** As illustrated in Figure 5(a), renderings from initial 3DGS contain blurry texture and artifacts. To address this, we leverage prompt-to-prompt editing with pre-trained text-to-image diffusion models to synthesize disocclusion areas, remove artifacts, and enhance geometry. Prompt-to-prompt editing (Hertz et al., 2022) modifies input images, which are described by the source prompt, to align with the target prompt while preserving structural content. Although typically used on real or diffusion-generated photos, we demonstrate its effectiveness for refining degraded satellite-trained 3DGS renders. We employ FlowEdit (Kulikov et al., 2024) with the pre-trained `FLUX.1 [dev]` diffusion model (Labs, 2024a), using prompt pairs that transform degenerate renders into high-quality imagery. Our prompts specifically describe the degraded features in original renders and specify the desired high-quality attributes in target prompts, see Section A.1 for prompts detail. As illustrated in Figure 5, this approach significantly improves the visual quality of renders, including sharper geometric details, enhanced texture richness, and physically coherent shadows, strengthening the 3DGS training dataset for more accurate reconstructions.

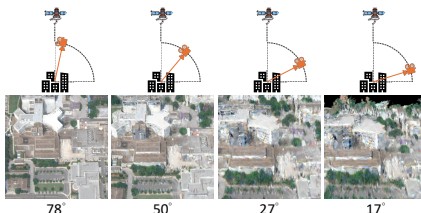

Figure 4: **The motivation of curriculum strategy.** Renderings of the initial 3D reconstruction from varied elevation angles reveal progressive degradation as the viewing angle decreases.

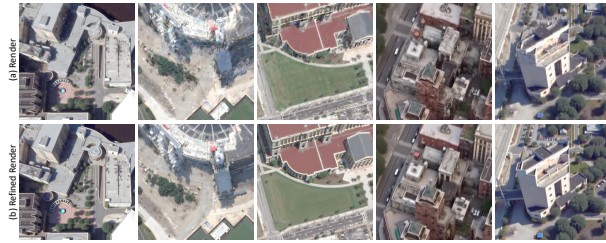

Figure 5: **Render refinement.** (a) Original 3DGS render with artifacts and blurry textures; (b) Refined result showing enhanced geometry and texture quality.

**Multiple diffusion samples.** While diffusion models effectively refine individual 3DGS renders, independently applying them across viewpoints introduces inconsistencies. Furthermore, 3DGS is well known to suffer from overfitting on single views, as pointed out by CoR-GS (Zhang et al., 2024b), causing artifacts when rendering from novel viewpoints.

Ideally, the optimal denoising diffusion process should produce a distribution where all views maintain synchronized 3D appearance. However, independent 2D denoising on each view does not preserve 3D consistency, resulting in a denoising trajectory distribution that is a super-set of the optimal trajectories. Selecting a single denoising trajectory from this expanded distribution has negligible probability of yielding the optimal 3D-consistent result, leading to the artifacts observed in Figure 9(c).

To mitigate this, we synthesize $N_s$ independently refined samples per view, effectively sampling multiple trajectories from the denoising distribution. During optimization, the photometric loss $\mathcal{L}_{\text{color}}$ implicitly averages over these $N_s$ samples. Rather than committing to a single potentially suboptimal denoising path, this approach allows the 3DGS optimization to find a consensus representation that balances fidelity to individual samples while promoting geometric coherence across views. Ablation studies (Section 4.2) and Figure 9(c) confirm that this strategy successfully balances detail preservation with structural coherence.

**Iterative dataset update.** Our curriculum-based Iterative Dataset Update (IDU), detailed in Algorithm 1, optimizes the 3DGS over $N_e$ episodes. In each episode, we render curriculum-guided views and refine them using FlowEdit (Kulikov et al., 2024) with specified prompts and strengths to generate a new training set. As the curriculum descends to lower altitudes, rendering quality steadily improves, particularly in previously occluded regions, as illustrated in Figure 6. We provide detailed parameters in Section A.1.

---

**Algorithm 1** 3DGS Refinement via Iterative Dataset Updates

---

**Input:** $N_e$: Number of episodes
**Input:** $N_v, N_s, N_p$: Number of views per point, samples per view and look-at points
**Input:** $\{P_i\}_{i=1}^{N_p}$: A set of $N_p$ target look-at points
**Input:** $\{R_i\}_{i=1}^{N_e}, \{E_i\}_{i=1}^{N_e}$: Decreasing sequences for radius and elevation with lengths of $N_e$
**Input:** $T_{\text{src}}, T_{\text{tgt}}, n_{\min}, n_{\max}$: FlowEdit parameters
**Input:** $G$: Initial 3DGS from satellite-view training
**Output:** $G'$: Refined 3DGS
1: $G' \leftarrow G$
2: **for** i = 1 to $N_e$ **do**
3:     radius $\leftarrow R_i$
4:     elevation $\leftarrow E_i$
5:     cam_views $\leftarrow$ ORBITVIEWS($\{P\}$, radius, elevation, $N_v$)        ▷ Generate $N_p \times N_v$ views
6:     render_views $\leftarrow$ RENDER($G'$, cam_views)        ▷ Render RGB images
7:     refine_views $\leftarrow$ FLOWEDITREFINE(render_views, $T_{\text{src}}, T_{\text{tgt}}, n_{\min}, n_{\max}, N_s$)        ▷ Refine
    renders using FlowEdit
8:     $G' \leftarrow$ TRAIN($G'$, refine_views)        ▷ Update 3DGS using refined views
9: **end for**
10: **return** $G'$

---

**Optimization.** For each episode $i$, we optimize the 3DGS using:

$$\mathcal{L}_{\text{IDU}}(G_{i-1}, \tilde{C}_i) = \mathcal{L}_{\text{color}} + \lambda_{\text{depth}}\mathcal{L}_{\text{depth}} , \quad (5)$$

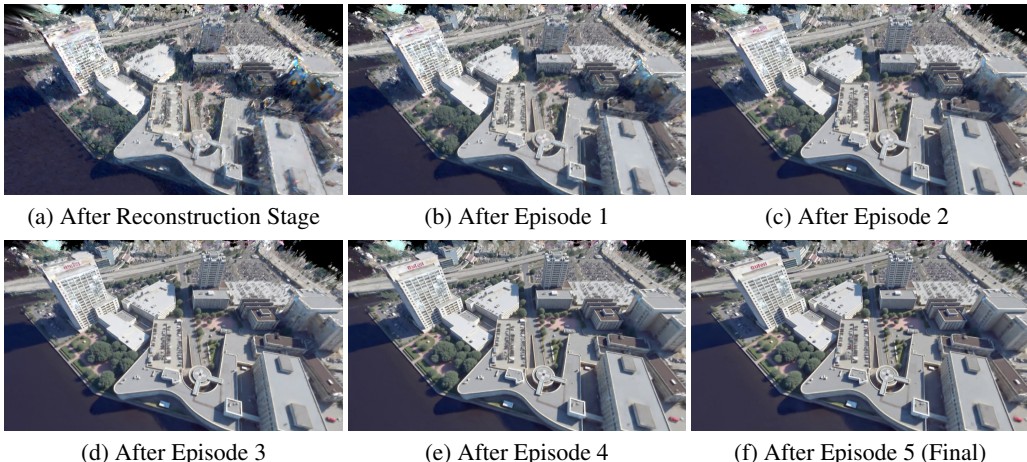

(a) After Reconstruction Stage     (b) After Episode 1     (c) After Episode 2

(d) After Episode 3     (e) After Episode 4     (f) After Episode 5 (Final)

Figure 6: **Visualization of progressive refinement.** This figure illustrates the step-by-step evolution of the synthesized 3D scene. Starting from the initial reconstruction state (a), the geometry and texture are progressively refined through successive stages of the iterative process (b-e), culminating in the final high-fidelity result (f).

where $G_{i-1}$ denotes the previous episode's 3DGS model, and $\tilde{C}_i$ are the current refined images. We provide more implementation details in Section A.1.

## 4 EXPERIMENTS

**Datasets.** We evaluate on high-resolution RGB satellite imagery from two sources. First, the 2019 IEEE GRSS Data Fusion Contest (DFC2019) (Le Saux et al., 2019) featuring WorldView-3 captures of Jacksonville, Florida (2048×2048 pixels, 35 cm/pixel resolution). Camera parameters and sparse points were generated using SatelliteSfM (Zhang et al., 2019). We evaluate on four standard AOIs: JAX_004, JAX_068, JAX_214, and JAX_260, following Sat-NeRF (Marí et al., 2022) and EOGS (Savant Aira et al., 2025) protocols. Second, for geographic diversity, we use the GoogleEarth dataset (Xie et al., 2024) (training data for CityDreamer (Xie et al., 2024) and GaussianCity (Xie et al., 2025b)) containing NYC scenes. We use four scenes (004, 010, 219, 336) with training views rendered at an 80° elevation to approximate satellite conditions. Google Earth Studio (GES) (Google, 2024) renders serve as ground truth for both datasets. See Section A.2 for more detail about datasets.

**Baselines.** Our method connects satellite-based 3D reconstruction and city generation, requiring baselines from both fields. For *satellite reconstruction*, we compare with Sat-NeRF (Marí et al., 2022) and EOGS (Savant Aira et al., 2025) on DFC2019 (they require RPC input unavailable in GoogleEarth), plus Mip-Splatting (Yu et al., 2024) (enhanced with our appearance modeling) and CoR-GS Zhang et al. (2024b) on both datasets.[1] For *city generation*, we compare with CityDreamer (Xie et al., 2024) and GaussianCity (Xie et al., 2025b) on GoogleEarth (their training dataset). We use official implementations with default settings. All experiments run on a single RTX A6000 GPU.

**Evaluation metrics.** We primarily use distribution-based metrics to quantify quality and diversity. We report $\text{FID}_{\text{CLIP}}$ (Kynkäänniemi et al., 2023) and CMMD (Jayasumana et al., 2024) that use the CLIP (Radford et al., 2021) backbone. This is based on their observations that the InceptionV3 (Szegedy et al., 2016) used in the classic FID (Heusel et al., 2017) and KID (Binkowski et al., 2018) is unsuitable for modern generative models. We complement these with user studies for perceptual quality assessment. We also report pixel-aligned metrics (PSNR (Huynh-Thu & Ghanbari, 2008), SSIM (Wang et al., 2004), LPIPS (Zhang et al., 2018)) as secondary references. While generally unsuitable for generative tasks, these metrics are meaningful for the Google Earth dataset, where all images come from the same consistent GES 3D representation, eliminating temporal variations.

---

[1]Many methods lack available code or models: Sat2Scene (Li et al., 2024d), Sat2Vid (Li et al., 2021), EO-NeRF (Marí et al., 2023), Sat-DN (Liu et al., 2025b), SatelliteRF (Zhou et al., 2024b), Sat-Mesh (Qu & Deng, 2023), CrossViewDiff (Li et al., 2024c), SkySplat (Huang et al., 2025), and others.

Table 1: **Quantitative comparison of different methods on DFC2019 (Le Saux et al., 2019).** The results show that our method consistently achieves the best performance, indicating superior perceptual fidelity compared to all baselines. Metrics are computed between renders from each method and reference frames from GES.

| Methods | Distribution Metrics | | Pixel-level Metrics* | | |
|---|---|---|---|---|---|
| | $FID_{CLIP} \downarrow$ | CMMD↓ | PSNR↑ | SSIM↑ | LPIPS↓ |
| *3D Reconstruction* | | | | | |
| Sat-NeRF (Marí et al., 2022) | 88.36 | 4.868 | 10.05 | 0.269 | 0.864 |
| EOGS (Savant Aira et al., 2025) | 87.74 | 5.286 | 7.26 | 0.168 | 0.959 |
| Mip-Splatting (Yu et al., 2024) | 87.19 | 5.405 | 11.89 | 0.318 | 0.819 |
| CoR-GS (Zhang et al., 2024b) | 89.03 | 5.241 | 11.55 | **0.350** | 0.948 |
| *Our Approach* | | | | | |
| Ours | **27.35** | **2.086** | **12.38** | 0.321 | **0.791** |

Table 2: **Quantitative comparison of different methods on GoogleEarth dataset (Xie et al., 2024).** The results show that our approach consistently achieves the best performance, indicating superior perceptual fidelity compared to all baselines. Metrics are computed between renders from each method and reference frames from GES.

| Methods | Distribution Metrics | | Pixel-level Metrics | | |
|---|---|---|---|---|---|
| | $FID_{CLIP} \downarrow$ | CMMD↓ | PSNR↑ | SSIM↑ | LPIPS↓ |
| *City Generation* | | | | | |
| CityDreamer (Xie et al., 2024) | 36.52 | 4.152 | 12.58 | 0.267 | 0.558 |
| GaussianCity (Xie et al., 2025b) | 28.73 | 2.917 | 13.41 | 0.291 | 0.541 |
| *3D Reconstruction* | | | | | |
| CoR-GS (Zhang et al., 2024b) | 27.32 | 3.752 | 12.85 | 0.291 | 0.455 |
| *Our Approach* | | | | | |
| Ours | **9.91** | **2.009** | **14.28** | **0.298** | **0.394** |

## 4.1 COMPARISONS WITH BASELINES

**Quantitative comparison.** We evaluate against both satellite reconstruction and city generation methods using distribution-based metrics. Evaluation images are created by dividing rendered frames into 144 patches ($512 \times 512$ pixels). For comparison in the DFC2019 dataset, we render GES reference videos at 17° elevation, extracting 30 frames per AOI (4,320 images total). For comparison in the GoogleEarth dataset, we use 45° elevation with 24 frames per scene (3,456 images total). We generate matching videos from all methods using identical camera parameters. Our method consistently outperforms all baselines across all metrics on both the DFC2019 and Google Earth datasets (Tables 1 and 2), demonstrating effective reconstruction across diverse urban environments.

**Qualitative comparison.** Figure 7(a) presents comparisons on the DFC2019 dataset against Sat-NeRF (Marí et al., 2022), EOGS (Savant Aira et al., 2025), and Mip-Splatting (Yu et al., 2024). All baselines exhibit significant distortions and blurry textures at lower viewpoints, while our baseline without IDU improves geometry but still shows floating artifacts and lacks facade detail. Our full approach achieves superior image quality. Figure 7(b) compares our approach on the GoogleEarth dataset against CityDreamer (Xie et al., 2024), GaussianCity (Xie et al., 2025b), and CoR-GS (Zhang et al., 2024b). While CityDreamer and GaussianCity generate plausible scenes, they produce oversimplified geometry and inaccurate textures, missing distinctive features such as the red pavement in scene 010 that our method correctly synthesizes. In contrast, our complete method achieves sharper building contours, enhanced texture fidelity, and reduced artifacts across both comparison scenarios. Notably, our approach successfully synthesizes plausible details for building facades occluded in the input satellite imagery and accurately reconstructs complex features including vegetation and multi-level architectures with finer surface details that better match the reference images. The visual quality approaches GES reference renders despite using only satellite imagery without ground-level data. Additional qualitative results are presented in Section A.2.

**User studies.** We conducted two comparative evaluations with 89 participants each: first, participants assessed the satellite input, GES reference video, Sat-NeRF, EOGS, CoR-GS, and our approach; second, participants compared the satellite input, GES reference video, CityDreamer, GaussianCity, CoR-GS, and our approach. Both studies evaluated geometric accuracy, spatial alignment, and overall quality, with full survey details in Section A.2. On the DFC2019 dataset, our method achieved dominant winrates of ≈97%/97%/97% vs. Sat-NeRF's ≈3%/3%/3%, while EOGS and CoR-GS achieved 0%/0%/0%. On the GoogleEarth dataset, our approach maintained a clear advantage with ≈90%/90%/92% winrates vs. CityDreamer's ≈4%/3%/3%, GaussianCity's ≈3%/3%/3%, and CoR-GS's ≈3%/4%/2%. These results consistently validate that our approach significantly outperforms all baselines under human perception across geometric accuracy, spatial alignment, and overall quality.

**Rendering efficiency.** Our method achieves 11 FPS on the modest NVIDIA T4 GPU, significantly outperforming CityDreamer's 0.18 FPS despite running on the far more powerful NVIDIA A100, which offers 5× the CUDA cores and 10× the memory bandwidth. GaussianCity reaches comparable speeds (10.72 FPS) but requires the high-end A100. Furthermore, our fused representation enables real-time rendering at 40 FPS on consumer hardware (MacBook Air M2), demonstrating that our method enables high-quality 3D urban navigation without specialized computing resources.

## 4.2 ABLATION STUDIES

We conduct ablation studies on the JAX_068 AOI.

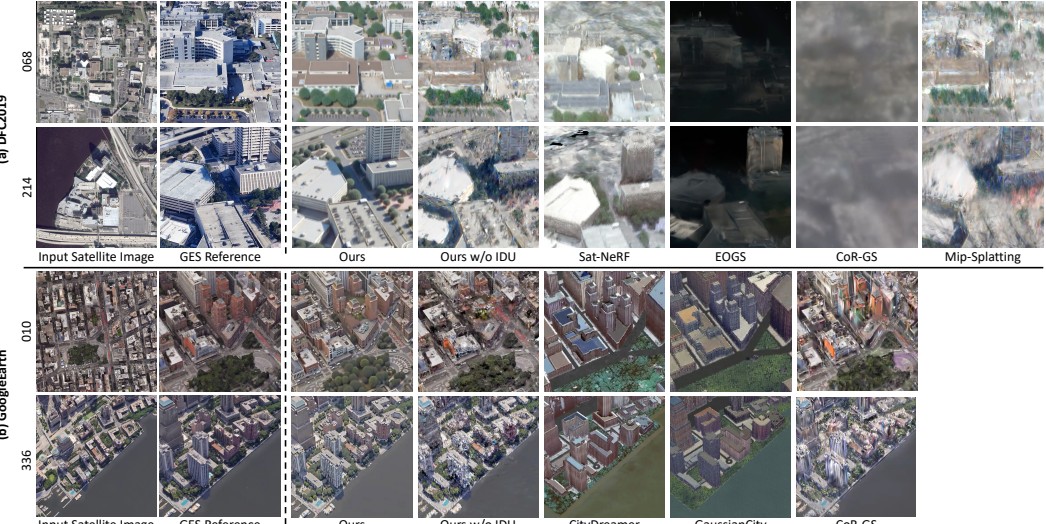

Figure 7: **Qualitative comparison on (a) DFC2019 and (b) GoogleEarth datasets.** The leftmost column shows one representative example of the input satellite images. Our method outperforms all baselines in geometric accuracy and texture quality in low-altitude novel views, demonstrating enhanced building geometry, detailed facades, and reduced floating artifacts. Notably, our approach correctly preserves distinctive features such as the red pavement in scene 010 that competing methods miss. Unlike CityDreamer (Xie et al., 2024) and GaussianCity (Xie et al., 2025b), our method operates directly on satellite imagery without requiring pixel-aligned semantic maps or height-fields, enabling synthesis of complex geometric structures that more closely match GES references.

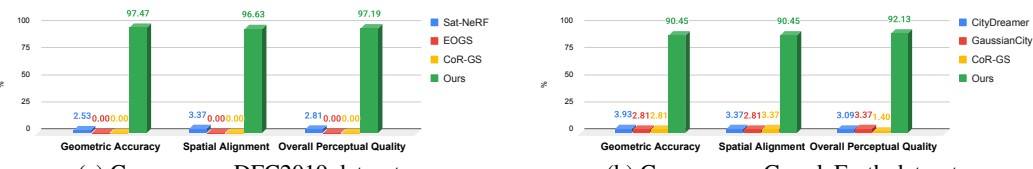

(a) Compare on DFC2019 dataset.    (b) Compare on GoogleEarth dataset.

Figure 8: **User study results.** Our method consistently outperforms Sat-NeRF (Marí et al., 2022), EOGS (Savant Aira et al., 2025), CoR-GS (Zhang et al., 2024b), CityDreamer (Xie et al., 2024) and GaussianCity (Xie et al., 2025b), achieving particularly high scores in geometric accuracy and overall perceptual quality. (a) details the comparison on the DFC2019 dataset (Le Saux et al., 2019), while subfigure (b) details the comparison on the GoogleEarth dataset (Xie et al., 2024).

**Ablation on the reconstruction stage.** We ablate appearance modeling, opacity regularization, and pseudo-camera depth supervision (see Table 3 and Figure 9). For this ablation, we evaluate at higher elevation angles to assess the quality of renders during the IDU process, rather than testing the final low-angle performance. Appearance modeling is crucial for multi-date convergence, opacity regularization removes floating artifacts (Figure 9(a)), and depth supervision flattens planar regions (Figure 9(b)). Together, they yield the lowest $\text{FID}_{\text{CLIP}}$/CMMD scores. Furthermore, we validate geometric accuracy using LiDAR data from the DFC2019 dataset (Le Saux et al., 2019). To quantify this, we unproject 3DGS depth renders into point clouds and rasterize them into Digital Surface Models (DSMs) for comparison. Our results show that both opacity regularization and pseudo-depth supervision improve geometric accuracy, with their combination achieving the lowest MAE/RMSE.

**Ablation on the synthesis stage.** We isolate two key factors: multi-sample diffusion and curriculum view progression. As Figure 9(c) shows, $N_s = 2$ achieves the optimal visual results. Although $N_s = 5$ yields the lowest CMMD, it requires a $1.5\times$ increase in training time with marginal returns in quality; thus, we adopt $N_s = 2$ for all experiments. Additionally, Figure 9(d) highlights that employing a curriculum strategy (vs. random views) effectively restores geometry in occluded areas, a benefit confirmed by Table 4. We further benchmark our refinement module against the SDEdit (Meng et al., 2022) baseline. As evident in Figure 9(e), SDEdit causes significant degradation, primarily due to its inability to hallucinate details while maintaining the structural integrity defined by the satellite imagery. Finally, we evaluate prompt sensitivity by utilizing generic context-free prompts. The negligible visual difference in Figure 9(e) confirms that our method is robust to prompt

Table 3: **Ablation on the reconstruction stage.** Appearance modeling secures convergence. Opacity regularization and depth supervision enhance visual fidelity and geometric accuracy.

| Components | | | Perceptual Metrics | | Geometric Metrics | |
|---|---|---|---|---|---|---|
| App. Mod. | Op. Reg. | Depth Sup. | $FID_{CLIP} \downarrow$ | $CMMD \downarrow$ | MAE (m)$\downarrow$ | RMSE (m)$\downarrow$ |
| ✗ | ✗ | ✗ | *Failed* | *Failed* | *Failed* | *Failed* |
| ✓ | ✗ | ✗ | 41.90 | 2.45 | 3.542 | 5.218 |
| ✓ | ✓ | ✗ | 39.95 | 2.40 | 2.980 | 4.527 |
| ✓ | ✓ | ✓ | **38.01** | **2.31** | **2.250** | **3.483** |

Table 4: **Ablation on the synthesis stage.** We evaluate sample counts ($N_s$), core components, and compare against baselines.

| Method Variation | $FID_{CLIP} \downarrow$ | $CMMD \downarrow$ | Time (h) |
|---|---|---|---|
| *Multiple Samples ($N_s$)* | | | |
| $N_s = 1$ | 34.11 | 3.19 | 3.44 |
| **Ours** ($N_s = 2$) | **28.35** | 2.88 | 6.37 |
| $N_s = 3$ | 28.64 | 2.77 | 7.19 |
| $N_s = 5$ | 29.17 | **2.68** | 9.80 |
| *Component Ablation* | | | |
| w/o Curriculum | 33.79 | 3.36 | - |
| w/ Context-free Pmt. | 30.78 | 2.98 | - |
| Replaced w/ SDEdit | 64.74 | 4.14 | - |

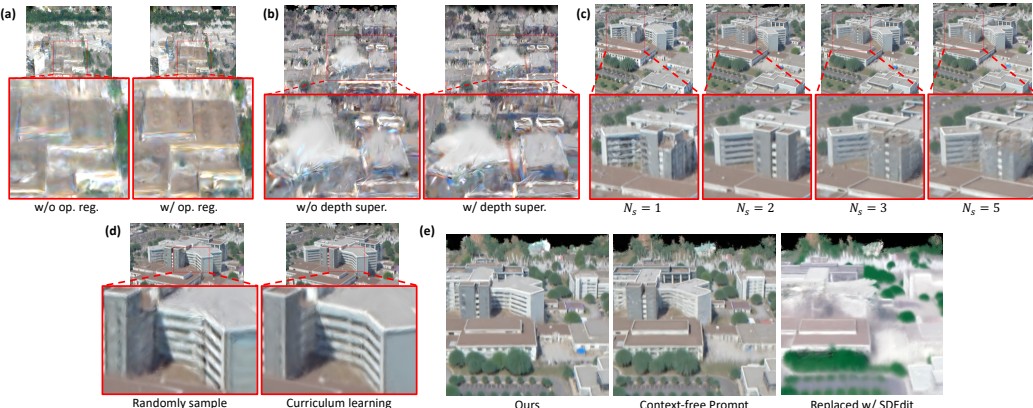

Figure 9: **Satellite-view training and IDU refinement ablation.** (a) Opacity regularization reduces floating artifacts and yields denser reconstructions. (b) Pseudo-camera depth supervision improves geometry in planar, texture-less areas like rooftops and roads. (c) Using multiple diffusion samples per view enhances texture consistency and reduces high-frequency geometric noise, $N_s = 2$ achieves the optimal visual results. (d) Curriculum learning progressively introduces challenging views, significantly improving geometric coherence in previously occluded regions compared to random sampling. (e) Refinement analysis: Using a generic context-free prompt results in a minor degradation of facade details but maintains structure, demonstrating robustness. In contrast, replacing our refinement method with SDEdit leads to a severe drop in quality, as standard noising-denoising struggles to hallucinate details while preserving the underlying geometry defined by the satellite imagery.

engineering and driven primarily by the diffusion model's internal priors. Please refer to Table 10 for the specific text prompts.

## 5 CONCLUSION

Skyfall-GS synthesizes real-time, immersive 3D urban scenes from multi-view satellite imagery, using 3D Gaussian Splatting and text-to-image diffusion models in a curriculum-based iterative refinement approach. Our method surpasses existing methods like Sat-NeRF, , EOGS, CityDreamer, and GaussianCity, effectively addressing challenges such as limited parallax, illumination variations, and occlusions. Future work includes scaling to larger environments and dynamic scenes.

**Limitations.** Our method requires significant computational resources, primarily due to the refinement process. The fixed heuristic camera trajectory creates blind spots in complex urban geometries, particularly in heavily occluded regions and scene boundaries. This results in artifacts and over-smoothed textures at extreme street-level perspectives. Additionally, our hybrid reconstruction-generation framework requires off-nadir satellite views. It cannot synthesize facades from purely top-down (nadir) imagery.

ETHICS STATEMENT

This work included a small-scale user study where anonymous participants were asked to compare our results with baselines through an online survey. No personally identifiable information was collected, and all responses were stored anonymously. Participation was entirely voluntary, and no risks were posed to participants. The study did not require institutional review board (IRB) approval under our institution's policies, as it involved only anonymous survey responses with minimal risk.

REPRODUCIBILITY STATEMENT

We have taken several steps to ensure the reproducibility of our work. Implementation details of our reconstruction and synthesis pipeline are provided in Section 3, including the architecture, loss functions, and optimization objectives. All hyperparameters, training schedules, and regularization terms are described in Section 3 and Section A.1. Details of datasets, splits, and evaluation protocols are described in Section 4 and Section A.2, with clear references to the publicly available DFC2019 dataset and GoogleEarth dataset. Details of the user study are described in Section A.2. We will release our source code upon acceptance to further support transparency and reproducibility.

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

# A  APPENDIX

This supplementary material provides additional details that complement our main paper. We include:

1. **Implementation Details**: This section details the pseudo-camera depth supervision strategy, 3DGS reconstruction parameters for satellite imagery, and the FlowEdit-based refinement process. We also provide a detailed breakdown of training time and memory consumption. Furthermore, we include a discussion on the validity of the RPC to perspective camera model approximation, including quantitative error analysis.

2. **Experiments Detail**: We provide dataset details, including training image counts and geographical coordinates for each Area of Interest (AOI), alongside the user study methodology and evaluation protocol.

3. **Additional Qualitative Results**: We present extended visual comparisons with state-of-the-art methods and results on four additional AOIs of the DFC2019 dataset. Furthermore, we provide visualizations of renders conditioned on varying per-image embeddings $e_j$ to illustrate temporal stability.

4. **Additional Experiments & Results**: We encompass a comprehensive set of new experiments, including: (i) synthesis results for complex, irregular geometries (e.g., castles and cathedrals) to demonstrate the framework's robustness; (ii) a sensitivity analysis of refinement text prompts;(iii) an episode-vs-coverage analysis to quantify the effectiveness of the curriculum strategy; and (iv) synthesized results with different random seed.

Additionally, we provide an interactive HTML visualization (available in the folder, `main.html`) that allows readers to explore our video results and compare reconstructions across different viewing conditions and scenes. This visualization enables direct comparison of our method's geometric accuracy, spatial alignment, and overall perceptual quality against baseline approaches and Google Earth Studio reference video.

We also provide example datasets via Zenodo, which can be accessed at this URL. However, due to storage limitations, we only provide training data for an AOI as an example. We plan to release the complete dataset upon acceptance.

## A.1  IMPLEMENTATION DETAILS

**Codebase.** Our method extends the Mip-Splatting (Yu et al., 2024) codebase with custom modules for satellite imagery processing and our curriculum-based IDU refinement pipeline.

**Pseudo camera depth supervision.** We sample cameras with varied azimuths and decreasing elevations, using random per-image embeddings. MoGe (Wang et al., 2024a) provides scale-invariant depth estimation. We sample 24 views every 10 iterations, with look-at points $(x, y, z)$, where $x, y \sim \mathcal{N}(0, 128)$ and $z = 0$, as illustrated in Figure 10. Camera azimuths are uniformly sampled between 0 and $2\pi$, while elevation angles and radii linearly decrease from $80°$ to $45°$ and 300 to 250 units, respectively. Rendered RGB images ($I_{\text{RGB}}$) are $1024 \times 1024$ pixels. We illustrate the 3DGS rendered RGB image $I_{\text{RGB}}$, scale-invariant depth $D_{\text{est}}$ estimated by MoGe (Wang et al., 2024a) and depth from 3DGS $D_{\text{GS}}$ in Figure 11.

**3DGS reconstruction from satellite imagery.** Our satellite-view optimization process runs for 30,000 iterations, with densification enabled between iterations 1,000 and 21,000. We modify several key parameters in the standard 3DGS implementation to address satellite imagery's unique challenges. First, to prevent undesirable Gaussian elongation artifacts common with overhead views, we reduce the scaling learning rate from 0.005 to 0.001. Second, we address sparsity issues of Gaussian points in close-up renderings by lowering the densification gradient threshold from 0.002 to 0.001, ensuring sufficient detail when viewed from ground level. Furthermore, we implement pruning of Gaussians with maximum covariance exceeding 20 to eliminate floating artifacts. The loss function weights are set to $\lambda_{\text{D-SSIM}} = 0.2$, $\lambda_{\text{op}} = 10$, and $\lambda_{\text{depth}} = 0.5$ for optimal reconstruction quality. For appearance modeling, we adopt the architecture from WildGaussians (Kulhanek et al., 2024), implementing an appearance MLP with 2 hidden layers (128 neurons each) and ReLU activation functions. The per-image and per-Gaussian embedding dimensions are set to 32 and 24 respectively, with learning rates of 0.001, 0.005, and 0.0005 for per-image embeddings $e_j$, per-Gaussian embeddings $g_i$, and the

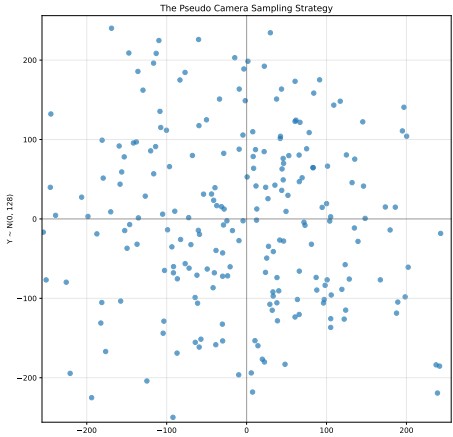

Figure 10: **The sampling strategy of pseudo camera.** In this example, we sample 240 points using the strategy.

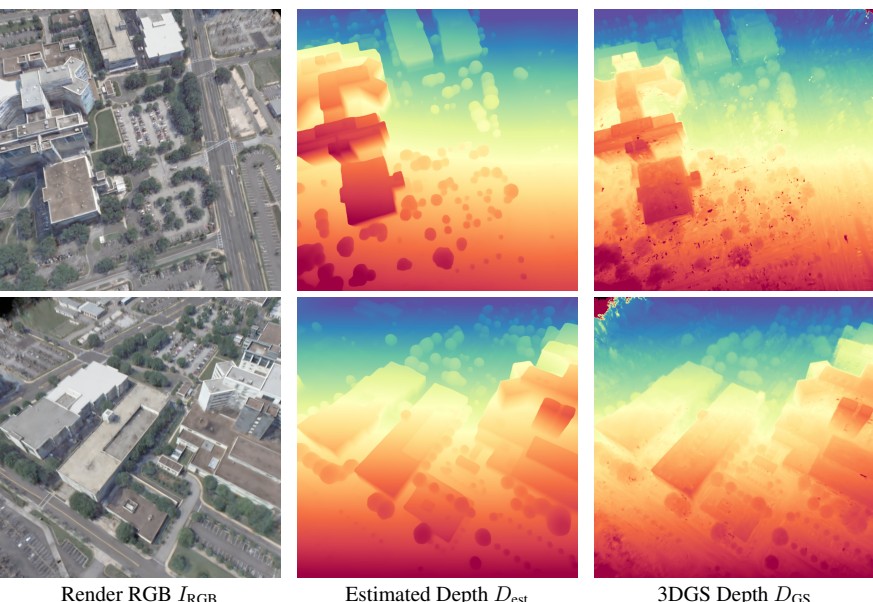

| Render RGB $I_{\text{RGB}}$ | Estimated Depth $D_{\text{est}}$ | 3DGS Depth $D_{\text{GS}}$ |

Figure 11: **Pseudo-cam Depth Supervision.** We use MoGe (Li et al., 2024c) to estimate the scale-invariant depth $D_{\text{est}}$ from the rendered RGB image $I_{\text{RGB}}$. The rightmost figures show the rasterized depth $D_{\text{GS}}$ from 3DGS.

appearance MLP $f$, respectively. The complete satellite-view training requires approximately 1 hour on a single NVIDIA RTX A6000 GPU.

**FlowEdit-based refinement.** We set FlowEdit noise parameters $n_{\min} = 4$ and $n_{\max} = 10$ to balance artifact removal with detail preservation. Our source prompt ("*Satellite image of an urban area with modern and older buildings, roads, green spaces. Some areas appear distorted, with blurring and warping artifacts.*") characterizes initial renders, while the target prompt ("*Clear satellite image of an urban area with sharp buildings, smooth edges, natural lighting, and well-defined textures.*") guides refinement. These parameters were determined through experimentation, with lower noise values preserving more original structure but removing fewer artifacts, and higher values creating

more significant changes but potentially altering underlying geometry. All other FlowEdit parameters use default values.

**Curriculum-based refinement details.** Our IDU process comprises $N_e = 5$ episodes of 10,000 iterations each, with densification through iteration 9,000. At the start of IDU, we randomly select and fix a single per-image appearance embedding $e_j$. Opacity regularization is disabled during IDU, as our curriculum naturally mitigates floating artifacts through multi-view consistency, enabling Gaussians to retain variable opacities beneficial for semi-transparent structures (Kerbl et al., 2023). For DFC2019 (Le Saux et al., 2019) dataset, we utilize $N_p = 9$ look-at points in a $3 \times 3$ grid (512 units wide, centered at origin), with $N_v = 6$ cameras per point and $N_s = 2$ samples per view. Camera elevations decrease from $85°$ to $45°$ and radii from 300 to 250 units across episodes. For GoogleEarth (Xie et al., 2024) dataset, we utilize $N_p = 16$ look-at point at origin, with $N_v = 6$ cameras per point and $N_s = 2$ samples per view. Camera elevations decrease from $85°$ to $45°$ and radius is fixed 600-unit across episodes. All training images are rendered at $2048 \times 2048$ resolution. Our training strategy samples 75% from refined images and 25% from original satellite images, this sampling strategy makes sure that the final 3DGS scene faithfully follows the semantic and layout in the input satellite imagery. The complete synthesizing stage requires approximately 6 hours on a single NVIDIA RTX A6000 GPU.

**Detail of training time.** All time measurements were conducted on the JAX_214 AOI using a single NVIDIA RTX A6000 (48GB) GPU. The total training time increases from approximately 1 hour 35 minutes for the baseline reconstruction (30K iterations) to 6 hours 45 minutes for the full pipeline. The majority of this additional cost is attributed to the Curriculum-based Iterative Dataset Update (IDU) process (5 episodes), which accounts for approximately 5 hours and 10 minutes combined. Specifically, a single IDU episode requires roughly 1 hour, where the computational load is split almost evenly between render refinement ($\sim$30 min) and 3DGS reconstruction update ($\sim$32 min), while the initial rendering step is negligible ($\sim$4 s). While this results in a total training time increase of approximately $4.3\times$, we view this as a justifiable offline investment to bypass physical data collection limitations.

**Detail of memory consumption.** We distinguish between peak memory and final memory. The peak memory usage reaches 46 GB during the synthesis stage, driven by the overhead of loading the diffusion model (FLUX.1) and temporary densification of Gaussians. However, the final training memory footprint is significantly lower (28.04 GB) as our method actively prunes redundant and low-opacity points. In terms of scene complexity, the refinement process densifies the scene by approximately 27%, increasing the Gaussian count from $\sim$1.65 million (reconstruction stage) to $\sim$2.1 million, specifically targeting the vertical facade geometry missing in the initial satellite reconstruction.

**Validity of RPC to perspective approximation.** We adopt the methodology proposed in SatelliteSfM (Zhang et al., 2019) to approximate the satellite linear pushbroom sensor as a perspective camera. This approximation relies on the "weak perspective" assumption, which holds valid when the satellite altitude ($Z$) is significantly larger than the depth variation within the scene ($\Delta Z$), i.e., $Z \gg \Delta Z$. Given that satellites orbit at distances of hundreds of kilometers while terrestrial depth variations are limited to a few hundred meters, the ratio $\Delta Z / Z$ remains negligible, allowing the geometry to converge to a perspective model. The approximation is achieved by generating a dense grid of 3D-2D correspondences using the rigorous RPC model and solving for a projection matrix $P$ via the Direct Linear Transformation (DLT) method, which is subsequently decomposed ($P = K[R|t]$) to recover camera parameters. Quantitative evaluations demonstrate that this process introduces negligible error: the average maximum forward projection error against the rigorous RPC model is only **0.126 pixels**, and the difference in triangulated 3D points is typically less than **5 cm**. Furthermore, this initialization allows Bundle Adjustment to achieve sub-pixel accuracy, with median reprojection errors recorded at **0.864 pixels**, confirming the suitability of this approximation for high-fidelity 3D reconstruction.

## A.2 Main Paper Experiments Detail & Results

**DFC2019 (Le Saux et al., 2019) dataset details.** The number of training images and geographical coordinates for each AOI is provided in Table 5. We also include four additional AOIs from

Table 5: **Number of training images and geographical coordinate per Area of Interest (AOI).** These AOIs correspond to standard evaluation scenarios established by previous works, ensuring consistent and fair comparisons with existing baselines (e.g., Sat-NeRF (Marí et al., 2022)).

| AOI | JAX_004 | JAX_068 | JAX_214 | JAX_260 |
|---|---|---|---|---|
| # of training image | 9 | 17 | 21 | 15 |
| Geographical coordinate | 81.70643°W, 30.35782°N | 81.66375°W, 30.34880°N | 81.66353°W, 30.31646°N | 81.66350°W, 30.31184°N |

Table 6: **Number of training images and geographical coordinates for additional AOIs.** We selected 4 additional AOIs with distinct characteristics: JAX_164 features a city hall building, JAX_175 contains an American football stadium, while the remaining two AOIs present other notable urban structures.

| AOI | JAX_164 | JAX_168 | JAX_175 | JAX_264 |
|---|---|---|---|---|
| # of training image | 20 | 21 | 21 | 21 |
| Geographical coordinate | 81.66362°W, 30.33032°N | 81.65297°W, 30.33037°N | 81.63696°W, 30.32583°N | 81.65285°W, 30.31189°N |

Jacksonville to demonstrate our method's robustness across varying scene characteristics. The number of training images and geographical coordinates for these additional AOIs is provided in Table 6. These additional AOIs feature distinct characteristics: one contains a city hall building (JAX_164), another includes an American football stadium (JAX_175), while the remaining two exhibit other notable urban features (JAX_168 and JAX_264).

**GoogleEarth (Xie et al., 2024) dataset details.** The GoogleEarth dataset, introduced by City-Dreamer (Xie et al., 2024), contains semantic maps, height fields and renders from Google Earth Studio (Google, 2024) of New York City. This dataset is used to train the generative model in CityDreamer (Xie et al., 2024) and GaussianCity (Xie et al., 2025b). We pick four AOIs which contain diverse city elements, including complex architectures (004), squares (010), resident area (219) and riverside (336). However, original GES renders provided in GoogleEarth dataset are rendered from a lower elevation angle, which is not similar to satellite imagery. Therefore, for each AOI, we render 60 images from GES using an orbit trajectory with $80°$ of elevation angle and 2219 of radius. These new renders serve as the input of our methods. The AOI ID, geographical coordinates, and the number of input images are detailed in Table 7.

**User study details.** We asked participants three specific questions and instructed them to select one video that best addressed each question:

1. **Geometric Accuracy**: "Which video's 3D structures (buildings, terrain, objects) more accurately represent the real-world geometry when compared to the ground truth video?"

2. **Spatial Alignment**: "Which video's layout and positioning of elements better matches the satellite imagery reference?"

3. **Overall Perceptual Quality**: "Considering all aspects (geometry, textures, lighting, consistency), which video presents a more convincing and high-quality 3D representation of the scene?"

For the user study on DFC2019 dataset, each participant viewed videos from Sat-NeRF (Marí et al., 2022), our method without IDU, and our complete method, alongside Google Earth Studio reference footage and the original satellite imagery. For the user study on the GoogleEarth dataset, each participant viewed videos from CityDreamer (Xie et al., 2024), GaussianCity (Xie et al., 2025b) and our complete method, alongside Google Earth Studio reference footage and the reference satellite imagery.

**Comparison details.** For quantitative comparisons with Sat-NeRF (Marí et al., 2022), Mip-Splatting (Yu et al., 2024) and our method without IDU refinement, we used consistent camera parameters across all methods: $17°$ elevation angle, 328-unit radius, and $20°$ field of view, with cameras targeting the AOI's origin. For comparisons with CityDreamer (Xie et al., 2024) and GaussianCity (Xie et al., 2025b), we use $45°$ elevation angle, 1067-unit radius, and $20°$ field of view, with cameras also targeting the AOI's origin. These parameters were selected to ensure equitable comparison with similar scene coverage across methods.

Table 7: **Number of training images and geographical coordinate per Area of Interest (AOI).** We pick 4 AOIs from the GoogleEarth (Xie et al., 2024) dataset, ensuring fair comparisons with existing baselines (e.g., CityDreamer (Xie et al., 2024) and GaussianCity (Xie et al., 2025b))

| AOI | 4WorldFinancialCtr (004) | 10UnionSquareE#5P (010) | 219E12thSt (219) | 336AlbanySt (336) |
|---|---|---|---|---|
| # of training image | 60 | 60 | 60 | 60 |
| Geographical coordinate | 74.01587°W, 40.71473°N | 73.98975°W, 40.73482°N | 73.98690°W, 40.73187°N | 74.01753°W, 40.71020°N |

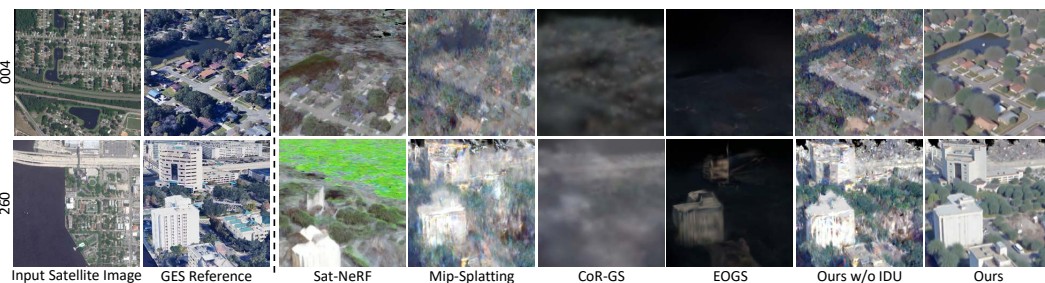

Figure 12: **Additional qualitative comparison on the DFC2019 dataset with Sat-NeRF (Marí et al., 2022), Mip-Splatting (Yu et al., 2024), CoR-GS (Zhang et al., 2024b), and EOGS (Savant Aira et al., 2025).** Our method significantly outperforms baseline approaches in both geometric accuracy and texture quality when rendering low-altitude novel views. Note the superior building geometry, facade details, and reduced floating artifacts in our final result.

**Per-scene quantitative comparison.** We provide per-scene quantitative comparison in Tables 8 and 9.

**Additional qualitative comparisons.** Due to space constraints in the main paper, we present additional qualitative comparison results in this supplementary material for scenes JAX_004 and JAX_260 from the DFC2019 dataset, and scenes 004 and 219 from the GoogleEarth dataset. Figure 12 shows orbital view comparisons with Sat-NeRF (Marí et al., 2022), Mip-Splatting (Yu et al., 2024), CoR-GS (Zhang et al., 2024b), and EOGS (Savant Aira et al., 2025), while Figure 13 presents city-scale view comparisons with CityDreamer (Xie et al., 2024), GaussianCity (Xie et al., 2025b), and CoR-GS (Zhang et al., 2024b). These additional results further demonstrate the consistent superiority of our method across diverse urban environments.

**Additional visual results.** We also provide qualitative results on four additional AOIs from Jacksonville to demonstrate our method's robustness across diverse urban environments. As shown in Figure 19 and Figure 20, these AOIs contain distinctive architectural features: JAX_004 showcases a residential area with mixed housing types and green spaces; JAX_164 features a prominent city hall building with its characteristic dome and symmetrical facade; JAX_175 encompasses an American football stadium with its distinctive oval structure and surrounding parking facilities; JAX_168 contains a commercial district with varied building heights and dense urban layout. Despite these varied urban typologies, our method successfully generates coherent three-dimensional renderings that preserve the spatial relationships and architectural features present in the satellite imagery. These additional results further validate the generalizability of our approach across diverse urban landscapes without requiring scene-specific parameter adjustments.

**Multi-date appearance variation.** The use of multi-date satellite imagery introduces a significant challenge, as images of the same location, when captured on different days, exhibit drastic variations in appearance. As shown in Figure 21, these differences can fundamentally alter the scene's geometry and texture. Effectively synthesizing novel views requires a model capable of intelligently disentangling the static 3D scene structure from these challenging, temporally-varying appearance factors.

A.3    ADDITIONAL EXPERIMENTS

**Qualitative results on complex geometries.** To demonstrate the robustness of our framework beyond standard city-block layouts, we evaluate our method on scenes featuring irregular and historically

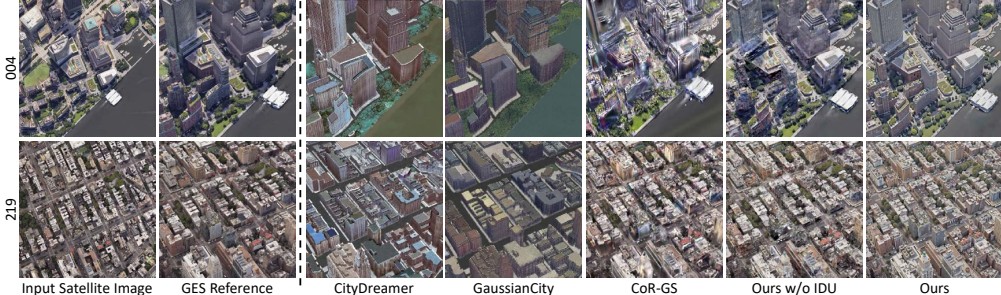

Figure 13: **Additional qualitative comparison on the GoogleEarth dataset with CityDreamer (Xie et al., 2024), GaussianCity (Xie et al., 2025b), and CoR-GS (Zhang et al., 2024b).** Our method is able to synthesize texture and geometry that is closer to the reference GES render.

Table 8: **Quantitative comparison on each AOI of DFC2019 (Le Saux et al., 2019).** Our method consistently outperforms baseline methods on distribution metrics and most pixel-level metrics, indicating superior image synthesis quality. Metrics are computed between renders from each method and reference frames from GES.

| Scene | Methods | Distribution Metrics | | Pixel-level Metrics* | | |
| | | FID$_{\text{CLIP}}$ ↓ | CMMD ↓ | PSNR ↑ | SSIM ↑ | LPIPS ↓ |
|---|---|---|---|---|---|---|
| JAX_004 | Sat-NeRF | 79.97 | 3.838 | 11.95 | 0.2290 | 0.8700 |
| | EOGS | 107.23 | 5.913 | 8.22 | 0.1271 | 1.0174 |
| | Mip-Splatting | 85.33 | 4.986 | 13.06 | 0.2412 | 0.8157 |
| | CoR-GS | 91.01 | 5.131 | 11.25 | **0.2554** | 0.9793 |
| | Ours | **24.45** | **1.474** | **12.90** | 0.2446 | **0.846** |
| JAX_068 | Sat-NeRF | 93.70 | 5.376 | 9.86 | 0.2607 | 0.8414 |
| | EOGS | 85.57 | 5.516 | 6.39 | 0.1593 | 0.9953 |
| | Mip-Splatting | 92.95 | 6.163 | 11.64 | 0.2900 | 0.8444 |
| | CoR-GS | 90.34 | 5.864 | 11.77 | **0.3230** | 1.0073 |
| | Ours | **28.35** | **2.845** | **11.79** | 0.2931 | **0.8210** |
| JAX_214 | Sat-NeRF | 90.76 | 5.376 | 8.97 | 0.2684 | 0.8394 |
| | EOGS | 71.02 | 4.342 | 7.40 | 0.2293 | 0.8883 |
| | Mip-Splatting | 82.04 | 5.088 | 11.23 | 0.3844 | 0.8048 |
| | CoR-GS | 86.33 | 5.258 | 11.66 | **0.4074** | 0.9079 |
| | Ours | **26.69** | **1.964** | **12.24** | 0.3881 | **0.7420** |
| JAX_260 | Sat-NeRF | 89.00 | 4.881 | 9.43 | 0.3172 | 0.9068 |
| | EOGS | 87.15 | 5.372 | 7.04 | 0.1574 | 0.9342 |
| | Mip-Splatting | 88.42 | 5.385 | 11.61 | 0.3579 | 0.8130 |
| | CoR-GS | 88.44 | 4.710 | 11.50 | **0.4162** | 0.8977 |
| | Ours | **29.83** | **2.076** | **12.59** | 0.3574 | **0.7540** |

significant architectures. As shown in Figure 20, we present synthesis results for **Neuschwanstein Castle** and **Wells Cathedral**. These scenes pose significant challenges due to their intricate non-Manhattan geometries, including sharp spires, varying elevations, and gothic architectural details. Despite these complexities, our method successfully disentangles the underlying geometry from the satellite input and hallucinates plausible high-frequency details for facades that are heavily occluded in the nadir views. This confirms that our hybrid reconstruction-generation approach is not limited to simple urban prisms but extends effectively to complex, free-form structures.

**Synthesis of bridges.** In addition to dense building clusters, we evaluate our method's performance on scenes with complex topological structures, such as bridges. Figure 16 illustrates renders of bridges in JAX_068, JAX_214 and JAX_175, a typically difficult case for standard photogrammetry due to the thin structural components. Our method successfully recovers the connectivity of the bridge span while synthesizing realistic water textures. The diffusion-based refinement effectively

Table 9: **Quantitative comparison with CityDreamer (Xie et al., 2024), GaussianCity (Xie et al., 2025b), CoR-GS (Zhang et al., 2024b) on each AOI of the GoogleEarth dataset (Xie et al., 2024).** The results show that our approach consistently achieves the best performance, indicating superior geometric and perceptual fidelity compared to all baselines. Metrics are computed between renders from each method and reference frames from GES.

| Scene | Methods | Distribution Metrics | | Pixel-level Metrics | | |
|---|---|---|---|---|---|---|
| | | $\text{FID}_{\text{CLIP}} \downarrow$ | CMMD$\downarrow$ | PSNR$\uparrow$ | SSIM$\uparrow$ | LPIPS$\downarrow$ |
| 004 | CityDreamer | 39.88 | 3.869 | 13.06 | 0.3519 | 0.5643 |
| | GaussianCity | 28.71 | 2.710 | 14.00 | 0.3786 | 0.5656 |
| | CoR-GS | 33.69 | 4.203 | 11.55 | 0.3440 | 0.6120 |
| | Ours | **10.43** | **2.491** | **15.09** | **0.3793** | **0.3978** |
| 010 | CityDreamer | 34.29 | 4.270 | 12.24 | 0.1387 | 0.5544 |
| | GaussianCity | 29.67 | 2.850 | 12.90 | 0.1661 | 0.5335 |
| | CoR-GS | 29.75 | 3.672 | 12.90 | **0.1807** | 0.4209 |
| | Ours | **11.03** | **1.631** | **13.58** | 0.1769 | **0.4073** |
| 219 | CityDreamer | 42.38 | 4.372 | 11.63 | 0.1344 | 0.5471 |
| | GaussianCity | 32.83 | 2.883 | 12.37 | 0.1676 | 0.5254 |
| | CoR-GS | 29.55 | 3.958 | 12.64 | **0.1792** | **0.3974** |
| | Ours | **7.83** | **2.635** | **13.12** | 0.1699 | 0.3975 |
| 336 | CityDreamer | 29.53 | 4.097 | 13.39 | 0.4431 | 0.5654 |
| | GaussianCity | 23.72 | 3.224 | 14.36 | 0.4533 | 0.5382 |
| | CoR-GS | 16.29 | 3.173 | 14.29 | 0.4592 | 0.3879 |
| | Ours | **10.36** | **1.279** | **15.32** | **0.4662** | **0.3719** |

regularizes the geometry, preventing the characteristic "melting" artifacts often observed in thin structures when using satellite-only reconstruction.

**Visualizing transient object handling via per-image embeddings.** A key challenge in multi-date satellite reconstruction is the handling of dynamic elements, such as moving vehicles and pedestrians, which can introduce ghosting artifacts. Our approach addresses this by learning per-image appearance embeddings $e_j$ that capture photometric variations specific to each capture date. As visualized in Figure 14, rendering the same viewpoint across 20 distinct appearance embeddings reveals that transient objects exhibit significant variability, appearing clearly in some embeddings while fading or vanishing in others. This qualitative evidence suggests that our appearance modeling effectively acts as a "sink" for transient data that does not align with the static 3D geometry. By absorbing these inconsistencies into the appearance code rather than the geometric parameters, the optimization naturally disentangles transient elements from the underlying static structure, ensuring a clean and consistent geometric reconstruction.

**Episode-vs-coverage analysis of curriculum strategy.** To quantify the effectiveness of the IDU module in revealing occluded regions, we present an Episode-vs-Coverage analysis (Figure 18). Since ground truth 3D geometry is unavailable for these satellite scenes, we use the final converged 3DGS model as a proxy for the total scene surface. We compute the cumulative coverage by optimizing a visibility attribute for every Gaussian point against the camera poses utilized in each episode. As shown in the figure, the coverage ratio steadily increases from $\sim$0.50 in Episode 1 to $\sim$0.75 in Episode 5. This consistent gain confirms that our curriculum strategy, which progressively lowers camera elevation from $85°$ to $45°$, successfully reveals and reconstructs vertical facade geometry that was initially occluded in the top-down satellite views. However, we acknowledge a limitation in this metric: because it calculates coverage based on reconstructed points, it cannot account for "true holes" (surface areas that were never generated at all because they were completely occluded from all sampled views). Future work could address this by dynamically sampling IDU cameras to target specific geometric uncertainties or detected holes.

**Stochastic appearance diversity.** To demonstrate the generative capacity of our hybrid framework, we evaluate the stochastic diversity of the synthesized textures in Figure 22. By varying the random seed during the diffusion refinement stage while maintaining the same geometric initialization, our

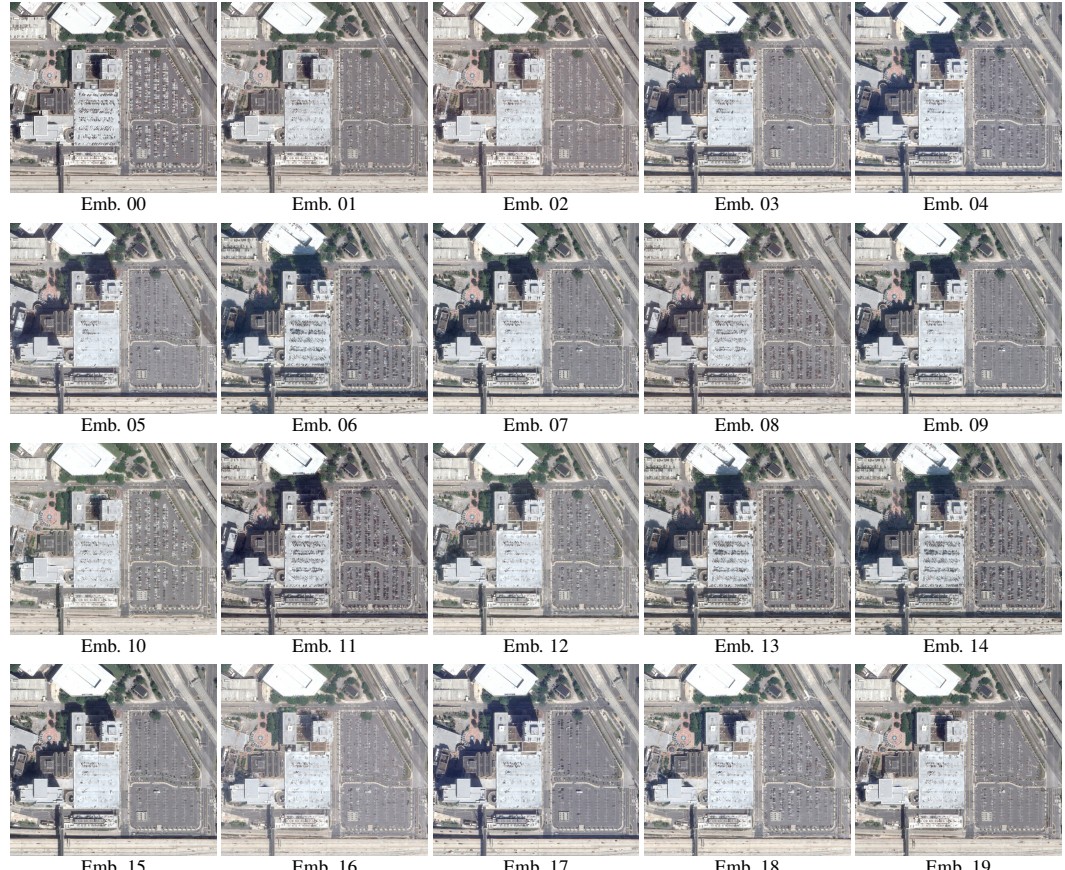

Figure 14: **Visualizing transient object handling via per-image embeddings.** We render the same viewpoint using 20 different learned appearance embeddings (Emb. 00–19). Observe that transient objects, such as the vehicles on the road, exhibit varying degrees of visibility across different embeddings (e.g., clearly visible in some, faded or absent in others), while the static building geometry remains consistent. This qualitatively demonstrates that our per-image appearance modeling effectively disentangles transient elements from the underlying static 3D structure, preventing dynamic artifacts from corrupting the geometric reconstruction.

method produces diverse yet plausible surface details for identical underlying structures, 5643]. As illustrated in the figure, detailed features such as the text on the red building signage vary distinctively (e.g., "Outeil" vs. "CUTAN"). Crucially, the macroscopic building footprint remains geometrically fixed, confirming that our framework successfully disentangles the reconstruction of physical geometry (grounded in satellite constraints) from the generative synthesis of high frequency appearance.

## A.4 LLM USAGE DISCLOSURE

Large language models (LLMs) were used to assist in improving the clarity and conciseness of the writing and in searching for related work. All technical ideas, algorithm designs, experiments, and analysis were conceived, implemented, and validated by the authors. The authors have carefully verified all content and take full responsibility for the correctness and integrity of this paper.

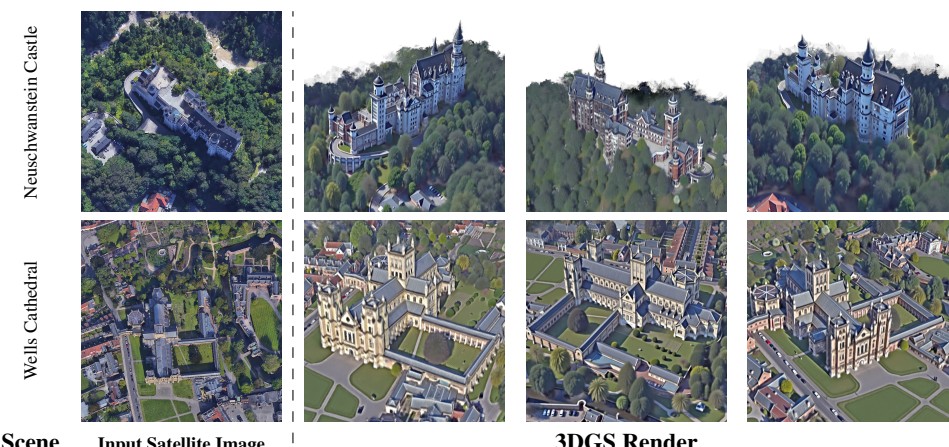

Figure 15: **Qualitative results on complex geometries.** Visualization of satellite image inputs and corresponding rendered frames. We demonstrate the model's capability on irregular historical architectures, including Neuschwanstein Castle and Wells Cathedral, showing the synthesis of complex geometry.

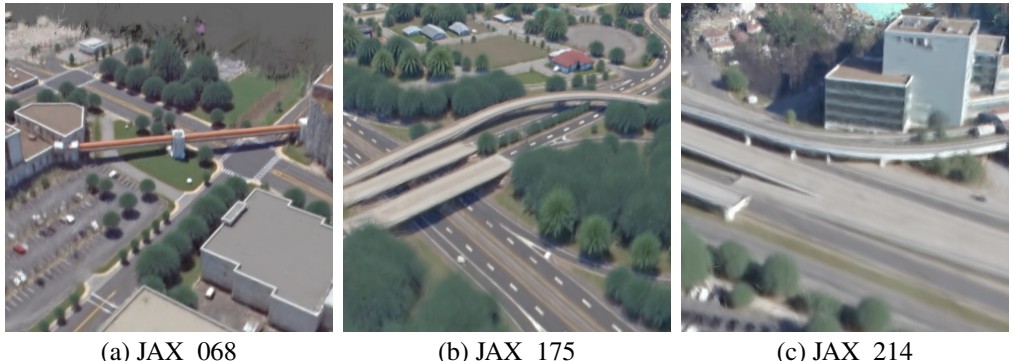

Figure 16: **Qualitative results for bridges.** We present the render results for bridges appears in JAX_068, JAX_214 and JAX_175, demonstrating the method's ability to handle complex topological structures and water surfaces that are typically challenging for standard reconstruction pipelines.

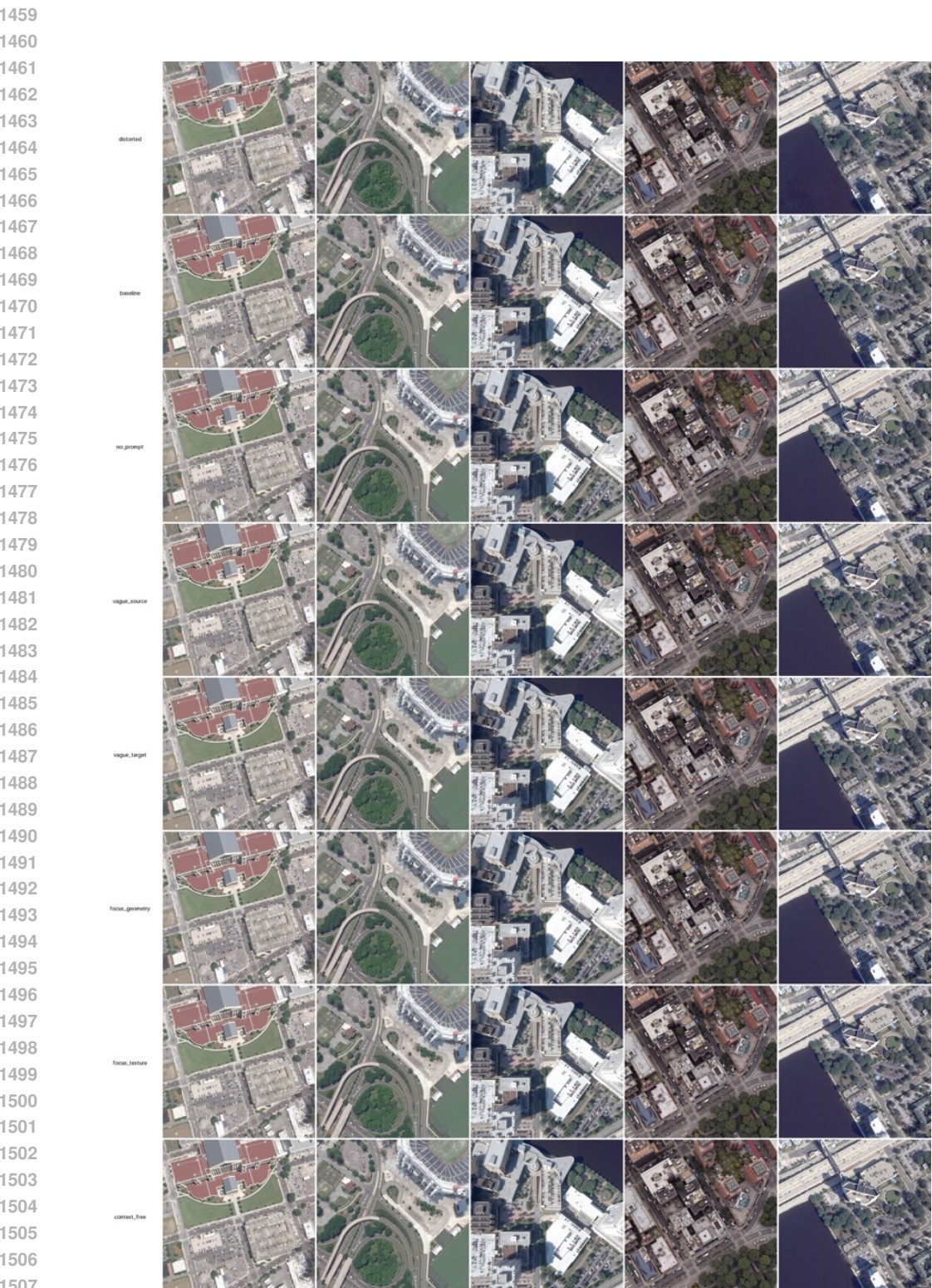

Figure 17: **Refine renders with different prompt strategies.**

Table 10: **List of text prompts used in sensitivity analysis.** We evaluate six different prompting strategies to test the robustness of our method.

| Strategy | Source Prompt ($P_{\text{src}}$) | Target Prompt ($P_{\text{tar}}$) |
|---|---|---|
| Baseline | Satellite image of an urban area with modern and older buildings, roads, green spaces. Some areas appear distorted, with blurring and warping artifacts. | Clear satellite image of an urban area with sharp buildings, smooth edges, natural lighting, and well-defined textures. |
| Vague Source | A blurry satellite image of an urban area. | Clear satellite image of an urban area with sharp buildings, smooth edges, natural lighting, and well-defined textures. |
| Vague Target | Satellite image of an urban area with modern and older buildings, roads, green spaces. Some areas appear distorted, with blurring and warping artifacts. | A clear satellite image of an urban area. |
| Focus Geometry | Satellite image of an urban area with modern and older buildings, roads, green spaces. Some areas appear distorted, with blurring and warping artifacts. | Clear satellite image of an urban area with geometrically precise buildings, flat rooftops, straight edges, and well-defined roads. |
| Focus Texture | Satellite image of an urban area with modern and older buildings, roads, green spaces. Some areas appear distorted, with blurring and warping artifacts. | Clear satellite image of an urban area with realistic, high-resolution textures, detailed facades, clear vegetation, and natural lighting. |
| Context Free | distorted, blurring, warping artifacts | clear, sharp, smooth edges, natural lighting, well-defined textures |

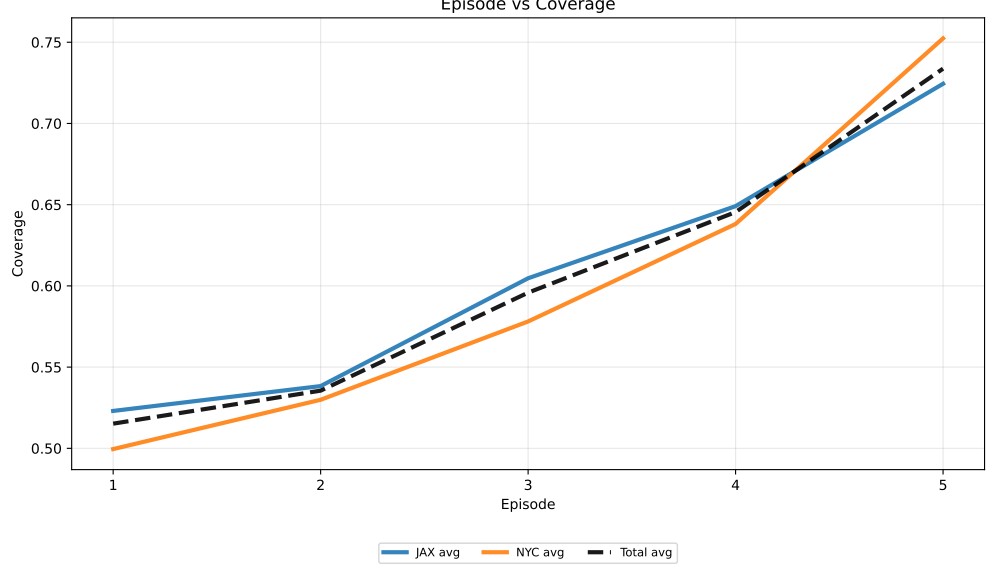

Figure 18: **Episode-vs-Coverage analysis.** The plot illustrates the cumulative surface coverage ratio increasing across refinement episodes. The curriculum-based strategy effectively exposes occluded regions, particularly vertical facades, as the camera elevation descends.

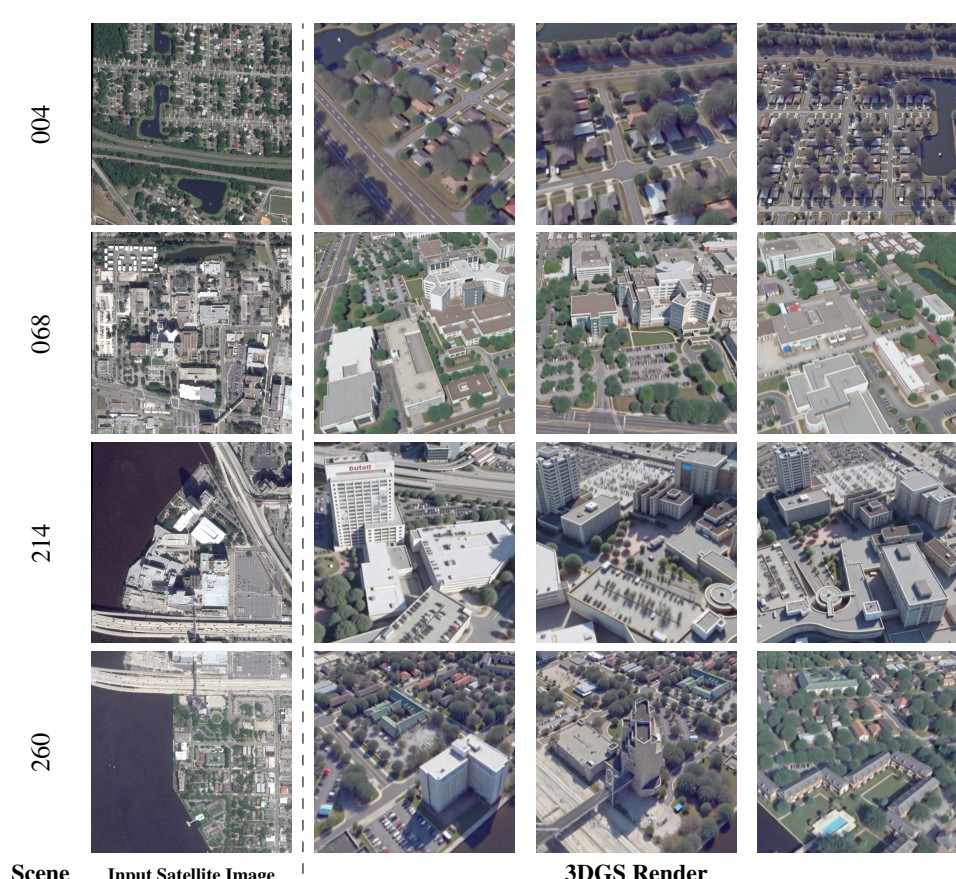

Figure 19: **Qualitative results across primary scenes.** Visualization of satellite image inputs and corresponding rendered frames for our four main AOIs.

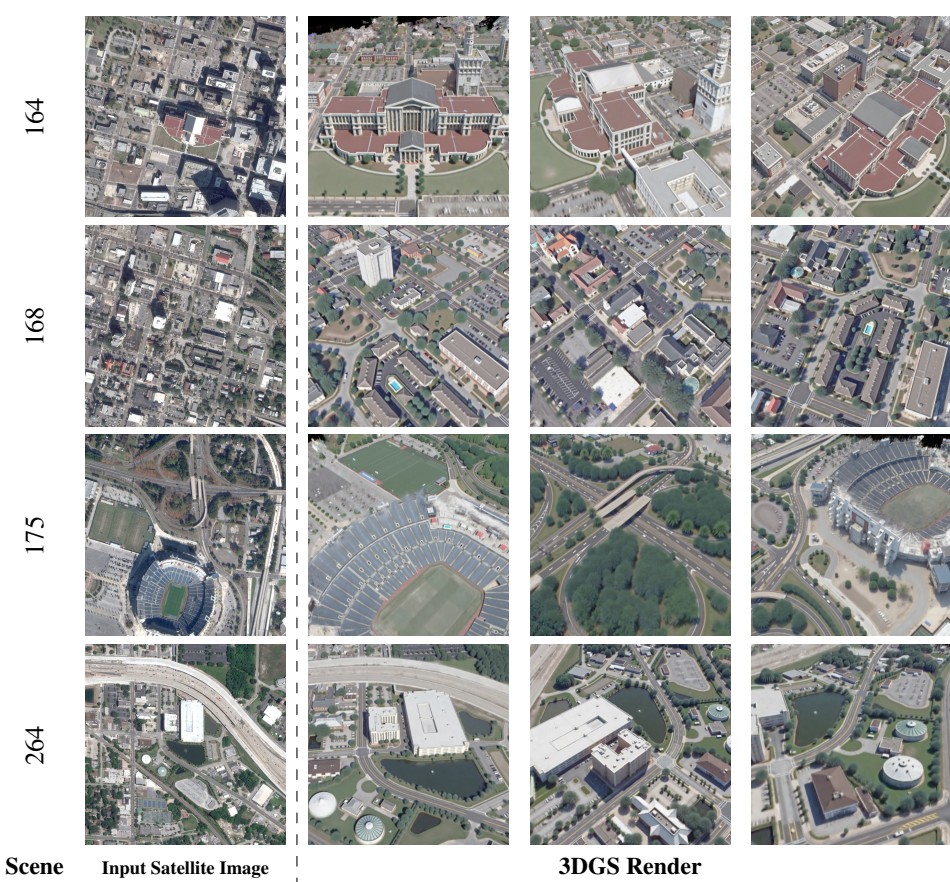

Figure 20: **Qualitative results across additional scenes.** Visualization of satellite image inputs and corresponding rendered frames for four additional AOIs with distinctive characteristics: JAX_164 features a city hall building, JAX_175 contains an American football stadium, while JAX_168 and JAX_264 present other notable urban structures.

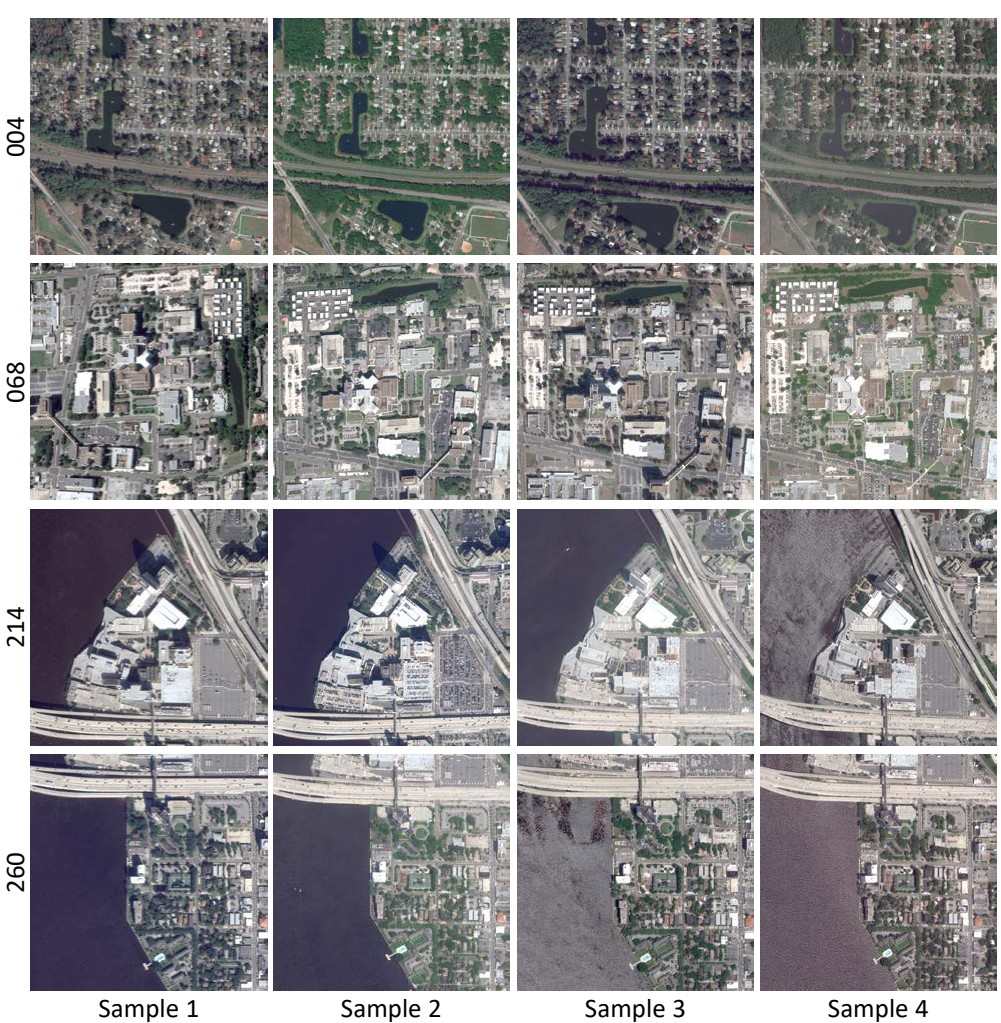

Figure 21: **Visualization of multi-date satellite imagery of the DFC2019 dataset.** Note the substantial shifts in appearance, including changes in illumination, cloud cover, and surface characteristics, which introduce challenges for consistent 3D reconstruction.

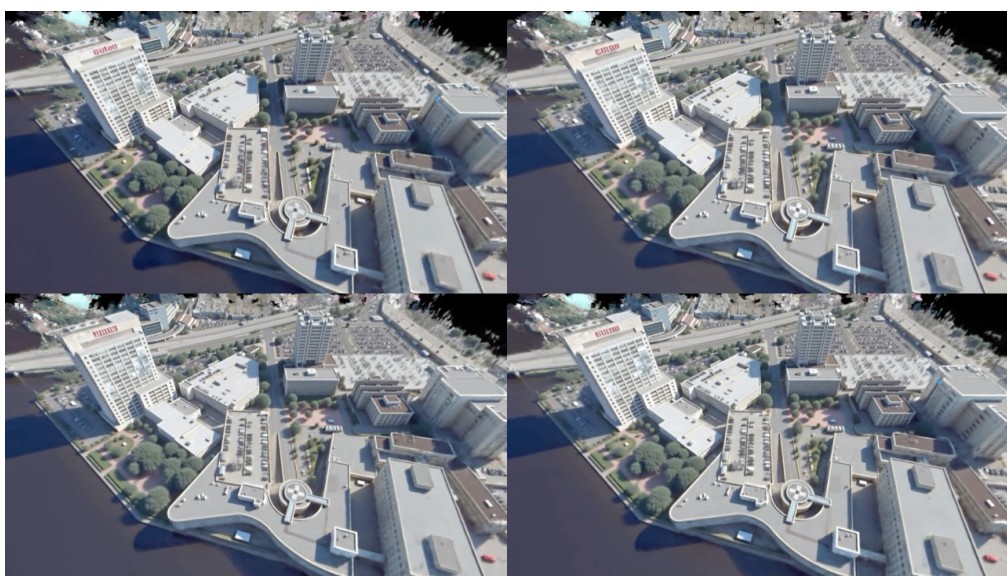

Figure 22: **Demonstration of stochastic appearance diversity while preserving geometric consistency.** Our method generates diverse plausible textures for identical underlying geometry across different random seeds. Notice how the red signage text on the building facade varies distinctively (e.g., "Outeil" vs. "CUTAN") while the building's structural footprint remains fixed, confirming that our framework successfully disentangles geometric reconstruction (grounded in satellite data) from generative appearance synthesis (variable via diffusion).

