# OpenReview forum: "Skyfall-GS: Synthesizing Immersive 3D Urban Scenes from Satellite Imagery"
_ICLR.cc/2026/Conference — Submitted to ICLR 2026_

### Official Review · Reviewer_HQFS · 2025-10-27

**Soundness:** 2
**Presentation:** 3
**Contribution:** 2
**Rating:** 4
**Confidence:** 5

**Summary:**

This work proposes a city-block scale scene synthesis framework, which aims to synthesis 3D urban scenes solely from satellite imagery, without relying on 3D or street-view data.
The framework first initializes coarse geometry from satellite imagery, then leverages off-the-shelf diffusion model to iteratively refine close-up appearances in a "sky fall" way.
This approach shows geometrically consistent, visually realistic, and detailed textures.

**Strengths:**

1. Clear motivation. This work presents an interesting approach to avoid using 3D data by reconstructing urban scenes directly from satellite imagery and refining them with 2D diffusion priors.
2. Significant problem setting. Immersive 3D City is essential for various applications in gaming, animations and virtual reality content.
3. Solid results. Quantitative metrics and user studies indicate that the proposed method outperforms existing state-of-the-art approaches. The ablation study is sufficient.

**Weaknesses:**

1. Ambiguous statement. From the current workflow, it is more like a reconstruction framework rather than a generative one. While the diffusion model improves visual quality for lower viewing angles, it's unclear how much diversity it can provide, given that the initial geometry is strongly determined in the reconstruction stage. The paper's claim "the first city-block scale 3D scene creation framework without costly 3D annotations" is therefore somewhat misleading. If this is a generative framework, diversity evaluation should be provided (although the visual results suggest potential overfitting).
2. Limited novelty. While the proposed framework is well-constructed, many of its technical components build upon existing methods. The reconstruction stage adopts techniques from SatelliteSfM[a], WildGaussians[b], and MoGe[c]. The iterative diffusion-based refinement for 3DGS is also explored in prior 3D scene generation methods (e.g., LucidDreamer[d], RealmDreamer[e], WonderWorld[f]).
3. Visual Quality.
   - The synthesized scenes contain noticeable blur, artifacts, and hollow areas, even in central regions (e.g., Supplementary/video_results/NYC_010/stage2_aligned.mp4, 00:01–00:03, red building in the middle). The proposed IDU module aims to gradually reveal occluded regions and refine them. However, it is unclear whether these artifacts and hollow arise from occlusion that is not revealed by the IDU technique, or whether the revealed regions cannot be effectively refined. In addiction, it would be helpful to include a metric such as the percentage change of revealed area (revealed / [revealed + occluded]).
   - The scene quality degrades significantly near the boundaries of synthesized region, which limits scalability to larger urban areas. Since the paper claims city-block-scale synthesis, it would be helpful to clarify how much of the synthesized region is actually usable or explorable. Is this degradation caused by the center-oriented curriculum refinement strategy?
4. Potentially biased demonstration. It is unclear whether the shown visual results overlap with the refinement iterations from synthesis stage, which may produces overly favorable visual results. Showing more dynamic camera viewpoints and path would provide better validation of the proposed method.

[a] Kai Zhang, Jin Sun, and Noah Snavely. Leveraging vision reconstruction pipelines for satellite imagery. ICCV Workshops, 2019.

[b] Jonas Kulhanek, Songyou Peng, Zuzana Kukelova, Marc Pollefeys, and Torsten Sattler. WildGaussians: 3D gaussian splatting in the wild. NeurIPS, 2024.

[c] Ruicheng Wang, Sicheng Xu, Cassie Dai, Jianfeng Xiang, Yu Deng, Xin Tong, and Jiaolong Yang. Moge: Unlocking accurate monocular geometry estimation for open-domain images with optimal training supervision. CVPR, 2025.

[d] Jaeyoung Chung, Suyoung Lee, Hyeongjin Nam, Jaerin Lee, Kyoung Mu Lee. LucidDreamer: Domain-free Generation of 3D Gaussian Splatting Scenes. arXiv 2311.13384, 2023.

[e] Jaidev Shriram, Alex Trevithick, Lingjie Liu, Ravi Ramamoorthi. RealmDreamer: Text-Driven 3D Scene Generation with Inpainting and Depth Diffusion. 3DV, 2025.

[f] Hong-Xing Yu and Haoyi Duan and Charles Herrmann and William T. Freeman and Jiajun Wu. WonderWorld: Interactive 3D Scene Generation from a Single Image. CVPR, 2025.

**Questions:**

Some more minor questions/discussion are in below:
1. How diverse are the synthesized results when the reconstruction fails to provide accurate geometry, especially when buildings are occluded in the satellite view or only top-down imagery is available (e.g., [Longitude 37.626, Latitude 55.752])?
2. How does the method handle complex or irregular geometries (e.g., bridges, castles[Longitude 10.749, Latitude 47.557], or Gothic buildings[Longitude -2.644, Latitude 51.21])? It would be impressive if the proposed method can successfully handle these challenging cases.
3. Have you experimented with concatenated satellite imagery for larger-area synthesis, and if so, how does performance scale?

---

> ### Author Response · Authors · 2025-11-23
> **Reply to Reviewer HQFS - Part 1**
>
> We thank the reviewer for their feedback on our framework's positioning and robustness; we address the concerns regarding complex geometries and occlusion below and have updated the manuscript to include these new validations.
>
> > **W1. Clarification on the statement**
>
> We thank the reviewer for this incisive observation. We agree the distinction between "reconstruction" and "generation" requires precise framing.
>
> We define our method as a **hybrid framework**:
> * **Reconstruction (Macro-Structure):** We enforce strict adherence to satellite-derived footprints and layouts. For digital twins, altering the layout would be a hallucination error.
> * **Generation (Micro-Texture):** High-frequency street-level details are **hallucinated** by the diffusion model to bridge the resolution gap, as satellite views lack sufficient angular density.
>
> Regarding diversity, while geometry is fixed, appearance is stochastic. We added Supplementary Figure 22 (L1743) showing diverse textures across seeds: for example, **red signage text** varies ("Outeil" vs. "CUTAN") while structural footprints remain identical. This confirms the disentanglement of fixed geometry from generative appearance.
>
> We have refined our claim to reflect this:
> * **Revised:** "...a novel **hybrid framework** that synthesizes immersive city-block scale 3D urban scenes by combining satellite reconstruction with diffusion refinement, eliminating the need for costly 3D annotations."
>
> This accurately reflects using generative models to solve an ill-posed reconstruction problem.
>
> > **W2. Limited novelty**
>
> We thank the reviewer for this critical question. While leveraging foundation models, our work is a **novel framework** addressing a significantly harder problem than general scene generation. Our technical novelty lies in two key areas:
>
> 1.  **Satellite-Specific Adaptations:** To address limited parallax, we introduce entropy-based opacity regularization and pseudo-camera depth supervision. These prevent the "floaters" and geometric collapse common when applying standard 3DGS to satellite data.
> 2.  **Domain-Specific Solution (Curriculum vs. Extrapolation):** Prior refinement works (LucidDreamer [a], RealmDreamer [b], WonderWorld [c]) cannot be naively applied. We introduce **Curriculum-based Iterative Dataset Update (IDU)** to bridge the domain gap:
>     * **Vertical Densification:** Unlike horizontal extrapolation in [a, c], Skyfall-GS performs *vertical densification*, hallucinating missing facades while strictly adhering to satellite footprints.
>     * **Curriculum Necessity:** Standard loops fail due to degraded initial ground renders. Our strategy, progressively lowering elevation ($85^\circ \to 45^\circ$), stabilizes diffusion guidance to plausibly reconstruct hidden facades.
>     * **Multi-Sample Consensus:** Unlike single-trajectory methods, we synthesize $N_s$ samples per view. This prevents overfitting to single hallucinations, forcing the optimization toward a 3D-consistent consensus.
>
> In summary, our work introduces a challenging new setting and the first effective solution via a non-trivial curriculum, distinct from the horizontal generation focus of prior art.

---

> ### Author Response · Authors · 2025-11-23
> **Reply to Reviewer HQFS - Part 2**
>
> > **W3(a). Artifacts in synthesized scenes**
>
> We thank the reviewer for identifying artifacts in NYC_010. We acknowledge that despite improvements, regions like the central red building exhibit blur or hollows.
>
> These hollows arise because the **IDU camera trajectory did not visually cover these regions**. Our fixed orbital curriculum ($85^\circ \to 45^\circ$) generally reveals facades, but complex occlusions create persistent "blind spots." If a region remains unseen by IDU pseudo-cameras, no Gaussians are densified, resulting in hollows.
>
> To quantify occlusion recovery, we present an **Episode vs. Coverage** analysis (Supp. Fig. 18; L1542). Lacking ground truth geometry, we use the final 3DGS model as a proxy, optimizing a visibility attribute (0-1) against episode camera poses. Results show coverage increasing from **~0.50 (Episode 1)** to **~0.75 (Episode 5)**, with "NYC avg" showing strong gains. This confirms our curriculum successfully reconstructs surfaces initially occluded in top-down views.
>
> We acknowledge a limitation: our metric uses *reconstructed* points and cannot account for "true holes" (surfaces never generated due to total occlusion). Future work could employ dynamic sampling to target these uncertainties. We have updated the manuscript to discuss this.
>
> > **W3(b). Degradation near the boundaries**
>
> We thank the reviewer for observing boundary degradation, which directly results from our center-oriented curriculum strategy. Our fixed $3 \times 3$ look-at grid concentrates refinement on the AOI center; boundaries receive minimal feedback due to infrequent, oblique views. Additionally, lower satellite parallax at boundaries yields weaker base reconstruction.
>
> We define the "explorable" zone within the look-at grid. To scale, synthesized blocks can be tiled with overlapping margins to discard degraded borders.
>
> > **W4. Potentially biased demonstration**
>
> We thank the reviewer for raising the point on evaluation fairness. We acknowledge the original video partially overlapped with IDU views.
>
> To eliminate bias and demonstrate robustness, we provide additional **free-flight trajectory videos** (Figshare: https://figshare.com/s/b6160a65f41f1a1244e). These trajectories follow smooth, dynamic paths never used during refinement, covering novel viewpoints and complex motions. This confirms our framework learns coherent, explorable 3D structure rather than overfitting to specific views.
>
> > **Q1. Robustness to occlusion and top-down imagery**
>
> We thank the reviewer for identifying this edge case. As a **hybrid reconstruction-generative pipeline**, our method relies on geometric cues (coarse structure, partial facades) from off-nadir views to anchor diffusion refinement.
>
> In strictly top-down (nadir) scenarios where facades are fully occluded, initial reconstruction fails, leaving no vertical surfaces for hallucination. Consequently, our method fails to generate accurate geometry. However, standard commercial acquisitions typically include off-nadir angles, providing the necessary geometric hints for success.
>
> > **Q2. Synthesizing complex and irregular geometry**
>
> We thank the reviewer for proposing these challenging test cases. We agree that irregular geometries like bridges and historical architecture represent significant stress tests.
>
> Regarding bridges, we provide qualitative results in Supplementary Figure 16 (L1438).
>
> To further demonstrate robustness, we conducted experiments on complex architectures. We provide **render videos** comparing our method against the Stage 1 baseline (3DGS with RPC model w/o IDU), showing that our hybrid approach effectively reconstructs spires, arches, and spans where pure reconstruction fails:
>
> * **Neuschwanstein Castle:** [https://figshare.com/s/adb017fa076d86ef097c](https://figshare.com/s/adb017fa076d86ef097c)
> * **Wells Cathedral:** [https://figshare.com/s/688325ddf6746c1feb14](https://figshare.com/s/688325ddf6746c1feb14)
> * **Qualitative Visualizations:** Supplementary Figure 15 (L1409)
>
> These results confirm the framework's generalizability beyond standard urban scenes.
>
> > **Q3. Scalability via concatenated satellite imagery**
>
> We thank the reviewer for the question on large-area synthesis. We agree that validating on concatenated imagery is crucial.
>
> We are currently combining neighboring AOIs (JAX_214 + JAX_260) to reconstruct a ~1km x 512m area. Our analysis focuses on boundary consistency and computational scaling; we will share these results shortly.
>
> ---
>
> [a] Jaeyoung Chung, Suyoung Lee, Hyeongjin Nam, Jaerin Lee, Kyoung Mu Lee. LucidDreamer: Domain-free Generation of 3D Gaussian Splatting Scenes. arXiv 2311.13384, 2023.
>
>
> [b] Jaidev Shriram, Alex Trevithick, Lingjie Liu, Ravi Ramamoorthi. RealmDreamer: Text-Driven 3D Scene Generation with Inpainting and Depth Diffusion. 3DV, 2025.
>
>
> [c] Hong-Xing Yu and Haoyi Duan and Charles Herrmann and William T. Freeman and Jiajun Wu. WonderWorld: Interactive 3D Scene Generation from a Single Image. CVPR, 2025.

---

> > ### Author Response · Authors · 2025-11-23
> > **Reply to Reviewer HQFS - Part 3**
> >
> > > **Q3. Scalability via concatenated satellite imagery**
> >
> > We have completed this experiment. We validated scalability by combining satellite imagery from JAX_214 and JAX_260 into a unified dataset covering a $\sim$1km $\times$ 512m area. We expanded the IDU look-at grid to **$6 \times 3$** while maintaining the standard 5-episode schedule, with each episode containing 10K iterations. Total training took **~9 hours** (1.5h reconstruction + 7.5h synthesis) on a single RTX A6000, yielding **~3.5M Gaussians** with a final training memory of $\sim$46GB. Qualitative results ([https://figshare.com/s/3f40b7683a4b840dfa7c](https://figshare.com/s/3f40b7683a4b840dfa7c)) demonstrate seamless boundary consistency (specifically around the highway) across the expanded area.

---

### Official Review · Reviewer_r2eL · 2025-10-28

**Soundness:** 3
**Presentation:** 3
**Contribution:** 3
**Rating:** 6
**Confidence:** 4

**Summary:**

This paper introduces Skyfall-GS, a novel method for generating city-scale 3D scenes from satellite imagery. The key contribution lies in a two-stage reconstruction pipeline: first, an initial coarse 3D scene is reconstructed using 3D Gaussian Splatting (3DGS); second, the result is refined with a pre-trained text-to-image diffusion model. The authors further design a curriculum-based iterative dataset update strategy to progressively enhance reconstruction quality. Experimental results demonstrate that the proposed approach achieves significant visual improvements over existing methods.

**Strengths:**

1. The manuscript is well-written, clear, and easy to follow.
2. The use of text-to-image diffusion models for refinement is conceptually sound.
3. The proposed method achieves strong visual and quantitative performance.

**Weaknesses:**

1. The paper employs pre-trained text-to-image diffusion models to iteratively refine reconstructed results. However, diffusion models are inherently stochastic, which means outputs can vary across different (or even identical) viewpoints. This randomness may lead to inconsistencies across views. Although the authors propose generating multiple samples to mitigate this issue, such an approach may cause over-smoothing, as the results could converge toward the mean of the distribution. A deeper discussion on this trade-off would strengthen the paper.
2. The authors use a prompt-to-prompt editing technique to guide image refinement by emphasizing geometric distortion and structural clarity between source and target images. It would be beneficial to further investigate the prompt design and analyze how different prompts influence refinement quality. Additionally, the authors could consider comparing this approach with noising-denoising image refinement pipelines explored in prior works [1].
3. To more clearly illustrate the effectiveness of the proposed refinement process, it would be helpful to include qualitative samples from multiple refinement stages, showing progressive improvements.
4. The paper discusses rendering efficiency but does not address training efficiency. Given that the proposed method involves a complex, multi-stage refinement pipeline, it would be valuable to provide an analysis or discussion of its training efficiency.

[1] Tang, Jiaxiang, et al. Dreamgaussian: Generative gaussian splatting for efficient 3d content creation. ICLR 2024.

**Questions:**

Please refer to weaknesses.

---

> ### Author Response · Authors · 2025-11-23
> **Reply to Reviewer r2eL - Part 1**
>
> We appreciate the reviewer's insightful suggestions regarding baselines and consistency; we address the trade-offs of diffusion sampling below and have incorporated the requested comparisons into the revised manuscript.
>
> > **W1 & Q1. Deeper discussion on multiple diffusion samples**
>
> We thank the reviewer for noting the trade-off between consistency and over-smoothing. While averaging multiple samples converges toward the mean, we leverage this as regularization in **satellite-to-ground synthesis**, tuning $N_s$ to balance geometric stability and texture retention.
>
> 1.  **Artifact Suppression:** High variance manifests as **geometric inconsistencies** (floaters) rather than valid texture. Single-sample denoising ($N_s=1$) overfits these artifacts, whereas multi-sample averaging filters out incoherent hallucinations (variance) to converge on the stable **geometric consensus** (mean).
> 2.  **Sample Count Ablation ($N_s$):** We extended our analysis to $N_s \in \{1, 2, 3, 5\}$ (Fig. 9(c); Table 4). Results indicate:
>     * **$N_s=1$:** Preserves sharpness but suffers from geometric noise.
>     * **$N_s=3, 5$:** Eliminates artifacts but causes over-smoothing.
>     * **$N_s=2$ (Ours):** Achieves the optimal balance, yielding the best $\text{FID}_{\text{CLIP}}$. While $N_s=5$ marginally improves CMMD, it increases training time by $1.5\times$.
>
> | Number of multiple samples | $\text{FID}_{\text{CLIP}} \downarrow$ | $\text{CMMD} \downarrow$ | Time (h) |
> | :--- | :---: | :---: | :---: |
> | $N_s = 1$ | 34.11 | 3.19 | 3.44 |
> | **Ours** ($N_s = 2$) | **28.35** | 2.88 | 6.37 |
> | $N_s = 3$ | 28.64 | 2.77 | 7.19 |
> | $N_s = 5$ | 29.17 | **2.68** | 9.80 |
>
> We conclude that $N_s=2$ minimizes geometric noise without succumbing to over-smoothing. This analysis has been added to the manuscript.
>
> > **W2(a) & Q2(a). The sensitivity of refinement text prompts**
>
> We appreciate the suggestion to investigate prompt sensitivity via two experiments.
>
> First, we tested five prompt strategies ranging from vague to specific (Supp. Table 10; L1514). Visual comparisons (Supp. Fig. 17; L1464) reveal consistent refinement quality, indicating the diffusion prior dominates the process regardless of specific wording.
>
> Second, we retrained JAX_068 using only generic "context-free" prompts (Source: "distorted, blurring..."; Target: "clear, sharp..."). While perceptually similar to the baseline (Fig. 9(e); L505), we observed a minor metric decline:
>
> | Methods | $\text{FID}_\text{CLIP}$ ↓ | $\text{CMMD}$ ↓ |
> | :--- | :--- | :--- |
> | Full Prompt | **28.35** | **2.88** |
> | Context-free Prompt | 30.78 | 2.98 |
>
> This confirms our framework is robust to prompt engineering, with detailed prompts offering only marginal statistical improvement. These results are included in the revised ablation studies.
>
> > **W2(b) & Q2(b). Comparison with noising-denoising pipeline**
>
> We thank the reviewer for the suggestion. To compare with **DreamGaussian** [a], we implemented a **noising-denoising (SDEdit [b])** baseline using **Stable Diffusion 2.1** on **JAX_068 AOI**, maintaining our curriculum schedule.
>
> SDEdit significantly degrades distribution metrics:
>
> | Methods | $\text{FID}_\text{CLIP}$ $\downarrow$ | $\text{CMMD}$ $\downarrow$ |
> | :--- | :--- | :--- |
> | **Ours** | **28.35** | **2.88** |
> | Noising-Denoising (SDEdit) | 64.74 | 4.14 |
>
> Qualitatively (Fig. 9(e); L505), SDEdit yields **blurry textures and inconsistent geometry**. High noise levels needed to correct artifacts cause hallucinations (e.g., changing roof styles, shifting roads) that contradict ground truth footprints, while lower noise fails to resolve inputs. Our method avoids this inherent conflict.
>
>
> > **W3 & Q3. Visualizing the progressive refinement process**
>
> We thank the reviewer for this constructive suggestion. We agree that visualizing scene evolution is crucial.
>
> We have added a comprehensive video (Figshare: [https://figshare.com/s/41979d4cb5bfdf34ef45](https://figshare.com/s/41979d4cb5bfdf34ef45)) and Figure 6 illustrating the refinement of JAX_214. The video traces the progression from noisy initial satellite reconstruction through iterative updates, demonstrating how our curriculum-based IDU pipeline stabilizes geometry and hallucinates high-fidelity textures. This confirms the gradual improvement in both geometric consistency and visual quality.

---

> ### Author Response · Authors · 2025-11-23
> **Reply to Reviewer r2eL - Part 2**
>
> > **W4 & Q4. Training efficiency**
>
> We thank the reviewer for the question on efficiency. We provide a cost breakdown using **JAX_214 AOI** on a single **NVIDIA RTX A6000 (48GB) GPU**.
>
> **1. Time Analysis:** Total training time increases from ~1.5h to ~6.75h, primarily driven by the iterative refinement loop.
>
> | Stage | Time (hours:minutes) | Relative Cost |
> | :--- | :--- | :--- |
> | **Baseline 3DGS Training** (30K iterations) | **1:35** | 1.0x |
> | **One IDU Episode** (10K iterations) | **~1:02** | |
> | $\quad\llcorner$ *Render generation* | ~0:00:04 | |
> | $\quad\llcorner$ *Render refinement* | 0:30 | |
> | $\quad\llcorner$ *3DGS reconstruction update* | 0:32 | |
> | **Total IDU Process** (5 episodes) | **~5:10** | |
> | **Complete Pipeline** (Baseline + IDU) | **~6:45** | **~4.3x** |
>
> **2. Memory Analysis:** Peak memory reaches **46 GB** due to diffusion model (`FLUX.1 [dev]`) overhead. Final memory remains lower due to active pruning and entropy-based regularization.
>
> | Stage | Peak Training Memory | Final Training Memory | 3DGS Count |
> | :--- | :--- | :--- | :--- |
> | **Reconstruction Stage** | 42.99 GB | **11.38 GB** | 1,655,919 |
> | **Synthesis Stage** | | | |
> | $\quad\llcorner$ *Render Refinement* | 45.00 GB | - | - |
> | $\quad\llcorner$ *Reconstruction Update* | **46.00 GB** | **28.04 GB** | **2,107,394** |
>
> **3. Trade-off:** While IDU increases compute $\sim$4.3x, this is a justifiable offline cost. Peak memory is transient; final memory remains compact. Refinement densifies the scene by **$\sim$27% ($\sim$450k points)**, targeting vertical facades while ensuring real-time rendering (40 FPS). This analysis is included in Supplementary (L1097, L1107).
>
> ---
>
> [a] Tang, Jiaxiang, et al. "DreamGaussian: Generative gaussian splatting for efficient 3d content creation." ICLR 2024.
>
> [b] Meng, Chenlin, et al. "SDEdit: Guided image synthesis and editing with stochastic differential equations." arXiv preprint arXiv:2108.01073 (2021).

---

### Official Review · Reviewer_1Pfy · 2025-10-30

**Soundness:** 3
**Presentation:** 3
**Contribution:** 3
**Rating:** 6
**Confidence:** 3

**Summary:**

The paper introduces Skyfall-GS, a framework for synthesizing immersive, block-scale 3D urban scenes from multi-view satellite imagery. It operates in two stages: Initial reconstruction using 3D Gaussian Splatting with appearance modeling, pseudo-camera depth supervision, and opacity regularization to build coarse geometry from satellites; Synthesis via curriculum-driven Iterative Dataset Update, leveraging pre-trained text-to-image diffusion models (e.g., FLUX.1) to iteratively refine renders from high to low elevation angles, filling occlusions (e.g., facades) and enhancing geometric sharpness and texture realism. No additional 3D annotations or street-level data are required.

**Strengths:**

1. Skyfall-GS demonstrates originality through a creative recombination of established tools in a novel domain.
2. The curriculum-driven Iterative Dataset Update (IDU) is a fresh problem formulation: treating 3DGS novel-view renders as noisy intermediates in a denoising diffusion process.
3. The reconstruction stage introduces targeted regularizers: entropy-based opacity pruning and Pearson-correlation depth supervision, which are directly address satellite-specific artifacts (floaters, limited parallax) with minimal overhead.

**Weaknesses:**

1. Some typo errors like 266L: this approach significant improves the visual quality ->this approach significantly improves the visual quality
2. Initial camera parameter approximation (RPC to perspective) may introduce errors in complex terrains, which needs some clarification.
3. Lack of handling of dynamic elements (e.g., vehicles, pedestrians); multi-date appearance modeling may leave transient artifacts.

**Questions:**

1. What is the exact spatial extent (m²) of evaluated scenes? Have you tested Skyfall-GS on contiguous multi-block areas (e.g., 1km×1km)?
2. In multi-date satellite sets, how are moving cars/trees handled? Can you show how to assess temporal stability
3. Is the elevation sequence {Ei} fixed or could it be conditioned on per-scene metrics

---

> ### Author Response · Authors · 2025-11-23
> **Reply to Reviewer 1Pfy**
>
> We thank the reviewer for identifying key technical details regarding camera models and scalability; we address these points below and have updated the manuscript with the new experimental results.
>
> > **W1. Some typos.**
>
> We thank the reviewer for spotting these typos. We have corrected them, including:
> 1. L056: "faces" $\to$ "face"
> 2. L268: "significant" $\to$ "significantly"
>
> > **W2. RPC to Perspective Approximation in Complex Terrains**
>
> We thank the reviewer for raising this point. We clarify that we adopt the **SatelliteSfM [a]** methodology to approximate RPC cameras as perspective cameras. This approximation is mathematically robust because it depends on the scene depth-to-altitude ratio ($Z \gg \Delta Z$) rather than absolute terrain variation.
>
> Since satellites orbit at hundreds of kilometers ($Z$) while scene variations ($\Delta Z$) are a few hundred meters, the ratio $\Delta Z / Z$ is negligible, allowing the pushbroom geometry to converge to a weak perspective model.
>
> The conversion involves a numerical fitting process:
> 1.  **Projection Fitting:** Generating 3D-2D correspondences via the rigorous RPC model within the AOI and solving for the projection matrix $P$ via Direct Linear Transformation (DLT).
> 2.  **Decomposition:** Decomposing $P = K[R|t]$ to recover intrinsics and extrinsics, where a non-zero skew in $K$ accounts for pushbroom motion.
>
> Empirically, SatelliteSfM [a] proves this introduces negligible error:
> * **Projection Fidelity:** Average max forward projection error against rigorous RPC is **0.126 pixels**.
> * **3D Consistency:** Triangulation difference is typically **< 5cm**.
> * **Refinement:** Sub-pixel median reprojection error (0.864px) after Bundle Adjustment.
>
> We included this validation and error analysis in Supplementary (L1115).
>
>
> > **W3 & Q2. Handling Dynamic Elements and Temporal Stability**
>
>
> We acknowledge the challenge of dynamic elements. Our framework mitigates their impact via two mechanisms:
>
> 1.  **Implicit Temporal Averaging:** Inconsistent appearance across multi-date captures causes 3DGS to treat transient objects as noise, assigning low opacity to prevent persistent geometric artifacts.
> 2.  **Appearance Embeddings:** Per-image embeddings ($e_j$) absorb transient data. New visualizations (Supp. Fig. 14; L1350) demonstrate that varying embeddings alters transient objects (e.g., vehicles appearing/vanishing) while preserving static geometry, confirming successful disentanglement.
>
> Qualitative results (Figs. 7, 19-20) show minimal artifacts. We discuss this implicit handling in Supplementary (L1325).
>
>
> > **Q1. Spatial extent and multi-block scalability**
>
>
> We thank the reviewer for the question on spatial scale. Our experiments cover city-block areas:
>
> * **DFC2019 (JAX):** 512m x 512m (~0.26 km²).
> * **GoogleEarth (NYC):** 640m diameter (~0.32 km²).
>
> For multi-block scalability, we are currently combining neighboring AOIs (JAX_214 + JAX_260) to reconstruct a ~1km x 512m area, focusing on boundary consistency and computational scaling.
>
> > **Q3. Elevation sequence configuration**
>
> The elevation sequence $\{E_i\}$ is a **tunable hyperparameter**. We empirically utilize a fixed schedule (85°, 75°, 65°, 55°, 45°) across both datasets, descending from near-satellite to drone perspectives over 5 episodes. This sequence was chosen to:
>
> 1.  **Ensure stable initialization:** 85° yields reasonable baseline renders from satellite-trained 3DGS (Fig. 4).
> 2.  **Manage difficulty:** 10° decrements provide gradual exposure to challenging viewpoints.
> 3.  **Ensure coverage:** The final 45° elevation captures target low-altitude drone perspectives.
>
> While this fixed schedule proves effective across diverse scenes (DFC2019, GoogleEarth), we acknowledge that adaptive sampling based on scene metrics (e.g., occlusion regions) could further improve quality. We leave this optimization for future work.
>
> ---
>
> [a] Kai Zhang, Jin Sun, and Noah Snavely. Leveraging vision reconstruction pipelines for satellite imagery. ICCV Workshops, 2019.

---

> > ### Author Response · Authors · 2025-11-23
> > **Reply to Reviewer 1Pfy - Part 2**
> >
> > > **Q1. Spatial extent and multi-block scalability**
> >
> > We have completed this experiment. We validated scalability by combining satellite imagery from JAX_214 and JAX_260 into a unified dataset covering a $\sim$1km $\times$ 512m area. We expanded the IDU look-at grid to **$6 \times 3$** while maintaining the standard 5-episode schedule, with each episode containing 10K iterations. Total training took **~9 hours** (1.5h reconstruction + 7.5h synthesis) on a single RTX A6000, yielding **~3.5M Gaussians** with a final training memory of $\sim$46GB. Qualitative results ([https://figshare.com/s/3f40b7683a4b840dfa7c](https://figshare.com/s/3f40b7683a4b840dfa7c)) demonstrate seamless boundary consistency (specifically around the highway) across the expanded area.

---

### Official Review · Reviewer_CcQH · 2025-10-31

**Soundness:** 3
**Presentation:** 3
**Contribution:** 3
**Rating:** 8
**Confidence:** 3

**Summary:**

This paper presents Skyfall-GS, a framework for synthesizing explorable 3D urban scenes directly from multi-view satellite imagery, without relying on ground-level data or 3D supervision. The pipeline has two stages: (1) initial 3D Gaussian Splatting (3DGS) reconstruction with appearance modeling, opacity regularization, and pseudo-depth supervision to address limited satellite parallax; and (2) a curriculum-based iterative dataset update (IDU) stage that refines renders using prompt-to-prompt diffusion editing. The model gradually transitions camera viewpoints from aerial to ground-level to recover occluded facades. Experiments on DFC2019 and GoogleEarth datasets show improvements over Sat-NeRF, CityDreamer, and GaussianCity in perceptual metrics (FID-CLIP, CMMD) and in user studies, with real-time rendering capability.

**Strengths:**

- The combination of 3DGS reconstruction and diffusion-based refinement is carefully engineered. The inclusion of appearance embeddings and pseudo-depth supervision for satellite data addresses unique challenges like illumination change and weak parallax.
- The elevation-progressive IDU strategy is original and intuitively sound, improving geometric coherence as camera angles descend from satellite to near-ground viewpoints.
- Quantitative results on two datasets, perceptual and pixel-based metrics, and two large user studies provide convincing validation. Ablations on all main components (opacity regularization, depth supervision, multi-sample diffusion, curriculum schedule) are thorough.

**Weaknesses:**

- While technically solid, the framework mainly integrates existing elements (3DGS + diffusion + curriculum learning). The novelty lies in applying these to satellite imagery rather than in introducing a fundamentally new algorithmic idea.
- Dependence on heavy computation. The iterative refinement process and multiple diffusion samples per view make the approach resource-intensive, somewhat at odds with the claimed scalability.

**Questions:**

- Could the authors quantify how the pseudo-depth supervision improves metric depth alignment (e.g., MAE vs. LiDAR) beyond perceptual scores?
- How sensitive is the refinement to the chosen text prompts? Would generic prompts (e.g., “clear buildings, realistic textures”) yield similar results?
- What are the memory and time costs of one complete IDU cycle relative to baseline 3DGS training?

---

> ### Author Response · Authors · 2025-11-23
> **Reply to Reviewer CcQH - Part 1**
>
> We appreciate the reviewer's constructive feedback on algorithmic novelty and efficiency; we address these points below and have incorporated the corresponding clarifications and cost analysis into the revised manuscript.
>
> > **W1. Limited novelty**
>
> We thank the reviewer for recognizing our framework's technical solidity. We appreciate the opportunity to clarify how our algorithmic contributions extend beyond applying foundational building blocks.
>
> Our work introduces **Curriculum-based Iterative Dataset Update (IDU)**, a novel strategy tailored to address satellite-to-ground reconstruction challenges unsolved by existing methods. Our contributions are threefold:
>
> 1.  **Algorithmic Novelty:** Unlike **horizontal extrapolation** methods (LucidDreamer [a], WonderWorld [b]), our vertical setting involves minimal overlap between satellite inputs and ground targets. Standard diffusion loops fail on the degraded initial renders (Fig. 2(a)). We propose **Curriculum-based IDU** to schedule camera elevation descent, enforcing a specific vertical hallucination order. This stabilizes the optimization landscape where standard methods diverge (Table 4; Fig. 9(d)).
> 2.  **Solving Generative Consensus:** Generative 3DGS often fails when independent 2D edits contradict in 3D. Unlike RealmDreamer [c] (text-prior reliance), we must adhere to strict ground footprints. We introduce a **multi-sample consensus mechanism** within the IDU loop, forcing 3DGS to average out inconsistent hallucinations into a geometrically plausible mean rather than overfitting single samples (Table 4; Fig. 9(c)).
> 3.  **Satellite-Specific Adaptations:** To address limited satellite parallax, we introduce entropy-based opacity regularization and pseudo-camera depth supervision. These are crucial to prevent floaters and geometric collapse, ensuring a viable foundation for subsequent refinement (Table 3; Fig. 9(a-b)).
>
> In summary, our IDU algorithm serves as a necessary technical bridge that enables existing tools to function in a domain where they otherwise fail.
>
> > **W2. Heavy computations & scability**
>
> We thank the reviewer for raising the concern regarding computational resources. While computationally intensive, our approach prioritizes **data and knowledge scalability** over runtime efficiency to address critical 3D modeling bottlenecks:
>
> 1.  **Data Scalability:** Unlike street/drone pipelines facing logistical barriers (regulations, manual labor), we utilize globally available satellite imagery. This removes the "human-in-the-loop" bottleneck, enabling scalable deployment across diverse regions.
> 2.  **Knowledge Scalability:** We trade computation to distill appearance priors from pre-trained diffusion models. Instead of collecting massive street-level datasets, we leverage foundation models as on-demand sources of urban knowledge. The cost is simply the mechanism to distill this knowledge into consistent 3D structures.
>
> This paradigm invests GPU hours to bypass expensive physical data collection, effectively offsetting the one-time computational cost per scene.

---

> ### Author Response · Authors · 2025-11-23
> **Reply to Reviewer CcQH - Part 2**
>
> > **Q1. Quantifying pseudo-depth supervision impact**
>
> We thank the reviewer for this question. We evaluated metric depth accuracy against LiDAR ground truth (JAX_068 AOI, DFC2019) to quantify the impact of pseudo-depth supervision:
>
> | Methods | MAE (m) $\downarrow$ | RMSE (m) $\downarrow$ |
> | :--- | :--- | :--- |
> | w/o Pseudo-depth Supervision | 2.980 | 4.527 |
> | **w/ Pseudo-depth Supervision** | **2.250** | **3.483** |
>
> Pseudo-depth supervision reduces errors by ~24%, significantly improving geometric accuracy. This confirms the method enhances both geometric fidelity and visual quality, complementing our perceptual metrics. These results are included in the revised manuscript (Table 3; L495).
>
> > **Q2. The sensitivity of refinement text prompts**
>
> We appreciate the suggestion to investigate prompt sensitivity via two experiments.
>
> First, we tested five prompt strategies ranging from vague to specific (Supp. Table 10; L1514). Visual comparisons (Supp. Fig. 17) reveal consistent refinement quality, indicating the diffusion prior dominates the process regardless of specific wording.
>
> Second, we retrained JAX_068 using only generic "context-free" prompts. While perceptually similar to the baseline (Fig. 9(e)), we observed a minor metric decline:
>
> | Methods | $\text{FID}_\text{CLIP}$ ↓ | $\text{CMMD}$ ↓ |
> | :--- | :--- | :--- |
> | Full Prompt | **28.35** | **2.88** |
> | Context-free Prompt | 30.78 | 2.98 |
>
> This confirms our framework is robust to prompt engineering, with detailed prompts offering only marginal statistical improvement. These results are included in the revised ablation studies.
>
> > **Q3. Memory & Time costs of IDU**
>
> We thank the reviewer for the question on efficiency. We provide a cost breakdown using **JAX_214 AOI** on a single **NVIDIA RTX A6000 (48GB) GPU**.
>
> **1. Time Analysis:** Total training time increases from ~1.5h to ~6.75h, primarily driven by the iterative refinement loop.
>
> | Stage | Time (hours:minutes) | Relative Cost |
> | :--- | :--- | :--- |
> | **Baseline 3DGS Training** (30K iterations) | **1:35** | 1.0x |
> | **One IDU Episode** (10K iterations) | **~1:02** | |
> | $\quad\llcorner$ *Render generation* | ~0:00:04 | |
> | $\quad\llcorner$ *Render refinement* | 0:30 | |
> | $\quad\llcorner$ *3DGS reconstruction update* | 0:32 | |
> | **Total IDU Process** (5 episodes) | **~5:10** | |
> | **Complete Pipeline** (Baseline + IDU) | **~6:45** | **~4.3x** |
>
> **2. Memory Analysis:** Peak memory reaches **46 GB** due to diffusion model (`FLUX.1 [dev]`) overhead. Final memory remains lower due to active pruning and entropy-based regularization.
>
> | Stage | Peak Training Memory | Final Training Memory | 3DGS Count |
> | :--- | :--- | :--- | :--- |
> | **Reconstruction Stage** | 42.99 GB | **11.38 GB** | 1,655,919 |
> | **Synthesis Stage** | | | |
> | $\quad\llcorner$ *Render Refinement* | 45.00 GB | - | - |
> | $\quad\llcorner$ *Reconstruction Update* | **46.00 GB** | **28.04 GB** | **2,107,394** |
>
> **3. Trade-off:** While IDU increases compute $\sim$4.3x, this is a justifiable offline cost. Peak memory is transient; final memory remains compact. Refinement densifies the scene by **$\sim$27% ($\sim$450k points)**, targeting vertical facades while ensuring real-time rendering (40 FPS). This analysis is included in Supplementary (L1097, L1107).
>
> ---
>
> [a] Jaeyoung Chung, Suyoung Lee, Hyeongjin Nam, Jaerin Lee, Kyoung Mu Lee. LucidDreamer: Domain-free Generation of 3D Gaussian Splatting Scenes. arXiv 2311.13384, 2023.
>
>
> [b] Hong-Xing Yu and Haoyi Duan and Charles Herrmann and William T. Freeman and Jiajun Wu. WonderWorld: Interactive 3D Scene Generation from a Single Image. CVPR, 2025.
>
>
> [c] Jaidev Shriram, Alex Trevithick, Lingjie Liu, Ravi Ramamoorthi. RealmDreamer: Text-Driven 3D Scene Generation with Inpainting and Depth Diffusion. 3DV, 2025.

---

### Author Response · Authors · 2025-11-23
**General Response**

We thank all reviewers for their constructive feedback! We are encouraged by the recognition of our method's **technical soundness (CcQH, 1Pfy, r2eL)**, **motivation (HQFS)**, and **performance (CcQH, r2eL, HQFS)**.

All revisions in the updated version are highlighted in red. Key updates include:

**New Experiments & Validations:**
* **Geometric verification:** Verified geometry against **LiDAR ground truth**, showing ~24% MAE reduction via pseudo-depth supervision (CcQH; Table 3).
* **Baselines & Ablations:** Extended sample count ($N_s$) analysis, added SDEdit comparison to prove structural stability, and performed prompt sensitivity analysis (r2eL, CcQH; Table 4, Fig. 9).
* **Robustness & Immersion:** Added renders of non-Manhattan scenes (Castles) and **free-flight videos** to demonstrate unbiased explorability (HQFS; Fig. 15, 16 & Figshare).
* **Scalability:** Completed **multi-block synthesis** merging two AOIs (~1km x 512m extent), demonstrating seamless boundary consistency (1Pfy, HQFS; Figshare), and added Episode-vs-Coverage analysis (HQFS; Fig. 18).

**Clarifications:**
* **Efficiency:** Detailed training time (~6.75h) and memory usage (CcQH, 1Pfy, r2eL; Supp. L1097).
* **Novelty:** Defined contribution as "vertical densification" within a **hybrid reconstruction-generation framework** (CcQH, HQFS).
* **Technical details:** Justified **RPC-to-Perspective approximation** and dynamic object handling via appearance embeddings (1Pfy; Supp. L1115 & L1325).

Thank you again for your time and effort. We look forward to any further discussions.

---

### Comment · Area_Chair_L2Xo · 2025-11-27

Dear Reviewers,

Thank you for your thoughtful evaluations of this submission. The authors have provided their responses and clarifications during the discussion phase. To ensure a well-informed final decision, I kindly encourage you to continue the discussion by reviewing the authors’ replies and adding any follow-up thoughts you may have.

If any of your original concerns remain unresolved, please feel free to raise them directly in the discussion thread.

Thank you again for your time and valuable contributions.

Best regards,

Area Chair

---

### Meta-Review · Area_Chair_4Rs6 · 2026-01-07

**Summary:**

Skyfall-GS synthesizes explorable 3D city-block scenes from multi-view satellite imagery with a two-stage pipeline. The paper reports better geometry/appearance metrics and LiDAR-aligned depth, plus real-time rendering. However, novelty is mainly integration, training is expensive, and visual artifacts/boundary limits remain.

**Reviewer Concerns:**

Addressed: added geometry validation, prompt sensitivity and SDEdit baseline, and evidence of seed-based texture diversity.

Still outstanding: limited core novelty, heavy dependence on a large diffusion prior and high compute, and remaining artifacts/boundary quality limits

**Reviewer Scores:**

CcQH: likely stays at 8 (key questions on depth, prompts, and cost are answered, but novelty/compute concerns remain). 1Pfy: likely stays at 6 (camera and scalability details are clarified, but overall impact is similar). r2eL: likely stays at 6 (added efficiency discussion and baselines help, but concerns about diffusion-side effects and complexity remain). HQFS: stays at 4, still below threshold.

---

### Decision · Program_Chairs · 2026-01-26

Reject